# Acoustic wave propagation in rivers: an experimental study

Thomas Geay[1], Ludovic Michel[1,2], Sébastien Zanker[2] and James Robert Rigby[3]

[1]Univ. Grenoble Alpes, CNRS, Grenoble INP, GIPSA-lab, 38000 Grenoble, France
[2]EDF, Division Technique Générale, 38000 Grenoble, France
[3]USDA-ARS National Sedimentation Laboratory, Oxford, Mississippi, USA

*Correspondence to*: Thomas Geay (th.geay@gmail.com)

**Abstract.** This research has been conducted to develop the use of Passive Acoustic Monitoring (PAM) in rivers, a surrogate method for bedload monitoring. PAM consists in measuring the underwater noise naturally generated by bedload particles when impacting the river bed. Monitored bedload acoustic signals depend on bedload characteristics (e.g. grain size

distribution, fluxes) but are also affected by the environment in which the acoustic waves are propagated. This study focuses on the determination of propagation effects in rivers. An experimental approach has been conducted in several streams to estimate acoustic propagation laws in field conditions. It is found that acoustic waves are differently propagated according to their frequency. As reported in other studies, acoustic waves are affected by the existence of a cutoff frequency in the kHz region. This cutoff frequency is inversely proportional to the water depth: larger water depth enables a better propagation of

the acoustic waves at low frequency. Above the cutoff frequency, attenuation coefficients are found to increase linearly with frequency. The power of bedload sounds is more attenuated at higher frequencies than at low frequencies which means that, above the cutoff frequency, sounds of big particles are better propagated than sounds of small particles. Finally, it is observed that attenuation coefficients are variable within 2 orders of magnitude from one river to another. Attenuation coefficients are compared to several characteristics of the river (e.g. bed slope, surface grain-size). It is found that acoustic waves are better

propagated in rivers characterised by smaller bed slopes. Bed roughness and the presence of air bubbles in the water column are suspected to constrain the attenuation of acoustic wave in rivers.

## 1 Introduction

### 1.1 Context of this study

Bedload transport monitoring is a challenging issue for river management. Geomorphological changes may be driven by

anthropogenic uses of rivers (e.g. hydroelectricity, sediment dredging, embankment, mining, land use changes) or to changes in available sediment loads related to extreme events or climate changes. Bedload transport is a dominant factor governing fluvial morphology but monitoring bedload transport is a difficult task. Direct sampling of bedload flux requires intensive field work, is difficult to accomplish during flood conditions. Additionally, direct sampling cannot provide automatic, continuous measurements during long periods, limited by the storage capacity and the need for operators. This is why the development of

surrogate (or indirect) methods has been studied in recent decades. The report of Gray et al. (2010) gives an overview of

available techniques. One of these methods concerns the use of bedload Self-Generated Noise (SGN). When bedload particles impact the river bed, an acoustic noise is created that propagates in the water column. Bedload SGN can be measured using hydrophones that are deployed in the river. Theoretical and experimental studies have shown that the acoustic power monitored by hydrophones can be related to bedload fluxes using power laws (Barton et al., 2010; Geay et al., 2017a; Johnson and Muir, 1969; Jonys, 1976; Marineau et al., 2016; Rigby et al., 2016a; Thorne, 2014). Some relations are also observed between bedload granulometry and spectral characteristics of SGN signals (Geay et al., 2017a; Thorne, 2014).

However, monitored signals are not only dependent on bedload SGN but also on propagation effects (Geay et al., 2017b; Rigby et al., 2016a). When propagating in rivers, bedload SGN suffers from geometrical spreading losses (Medwin, 2005), multiple diffractions on rough boundaries (Wren et al., 2015) or from other attenuation processes, for example related to the occurrence of suspended load (Richards et al., 1996). Therefore, acoustic waves are modified by the environment along their propagation paths, from noise sources to hydrophone measurements. It has been shown that the river could be modeled as an acoustic wave guide where acoustic waves are partially trapped between the water surface and the river bed (Geay et al., 2017b). The occurrence of a cutoff frequency (related to the Pekeris waveguide) has been observed in field experiments (Geay et al., 2017b; Lugli and Fine, 2007) and reported in a theoretical review (Rigby et al., 2016b). A laboratory study focused on the role of river bed roughness as a source of attenuation process (Wren et al., 2015): an increase of 4 dB with an increasing bed roughness of 20 mm has been observed. There is comparatively little literature in the range of frequencies of interest (i.e. 0.1 to 100 kHz) and none of these studies have done specific experiments to define acoustic propagation laws in field experiments. For this reason, we designed a new protocol enabling the determination of propagation laws in rivers. These experiments result in experimental laws that are useful for building direct or inverse models, which is necessary to analyze bedload SGN signals. For example, it could be used to better understand the measurement range of a hydrophone in a river, a question which remains unknown.

The next section of the manuscript relates a simple theoretical framework that is used to analyze field data. The second part of this paper describes the protocol which is based on emitting a known signal with an active source (i.e. an underwater speaker) and on measuring this same signal at several distances from the source. The third part is related to the application of this protocol in a set of rivers that have different morphology (e.g. water depth, slope, flow velocities, bed roughness). The variation of propagation properties is observed from one river to another and related to river characteristics.

## 1.2 Theoretical framework

Acoustic measurements are in part determined by the ability of the environment to propagate sounds. In this section, an acoustic theory is proposed to model the loss of acoustic power with the distance of propagation. At a first stage, without attenuation processes, the monitored power ($P$ - $\mu Pa^2$) of a sound source decreases with distance from the point source as the energy is spreads in space:

$$P(r) = P_{@1m}\, G(r) \tag{1}$$

Where $r$ (m) is the distance from the source to the sensor, $P_{@1m}$ ($\mu Pa^2$ @1m) is the initial power of the sound source monitored at 1 meter in a free field and $G$ is a function depicting geometrical spreading. The geometry of the river is simplified as a rectangular channel with a uniform water depth, denoted $h$. For underwater acoustic waves propagating in a river, the medium is bounded by the water surface and the river bed. The effect of river banks is not explicitly considered in this study. It is assumed that banks act as efficient sound absorbers. At the upper and lower interfaces, reflection coefficients are variables, depending on the geo-acoustic parameters of the river bed (Geay et al., 2017b) and on the roughness of the interfaces (Wren et al., 2015). Two extreme cases can be assumed. First, when the interfaces are perfectly reverberant, acoustic waves are totally trapped into the water column and acoustic waves propagate in a cylindrical way. For large distance of propagation (i.e. $r > h$):

$$G(r) = \frac{2}{rh} \tag{2}$$

Secondly, when the interfaces are highly absorbing (as in an anechoic chamber), acoustic waves propagate in a spherical mode as in a free space:

$$G(r) = \frac{1}{r^2} \tag{3}$$

In the following, both propagation laws (spherical or cylindrical) will be tested to fit field data.

Acoustic waves not only suffer from geometrical spreading but also from losses from other processes that attenuate sounds like absorption or scattering effects. As stated in ocean studies, it is not really possible to distinguish both effects in field experiments (Jensen et al., 2011). In this study, we propose to quantify these effects in a single exponential term as written is the following equation:

$$P(r) = P_{@1m}\, G(r)\, e^{-2\alpha r} \tag{4}$$

Where $\alpha$ is a coefficient of attenuation (nepers/m), $\alpha > 0$.

The attenuation of acoustic waves is a process which is frequency dependent. That is why it is common to express the coefficient of attenuation as a function of wavelength (Jensen et al., 2011), denoted here $\alpha_\lambda$ (nepers):

$$\alpha_\lambda = \alpha\lambda = \alpha\frac{c}{f} \tag{5}$$

where $\lambda$ is the wavelength (m), c is the velocity of sound in water (m/s) and f the frequency (Hz).

The goal of this study is to experimentally determine the values of the attenuation coefficients for acoustic waves in rivers, for frequencies between 1 kHz-100 kHz. This range of frequency corresponds to the expected range of frequencies generated by bedload self-generated noise of particles size between $10^{-1}$ and $10^{-3}$ m (Thorne, 2014).

## 2 Experimental Setup

An experimental set-up was designed to measure the loss of acoustic power with distance of propagation in natural streams. A controlled sound source emits a known signal at a fixed position on the river bed and this same signal is monitored with a hydrophone, at several distances from the point of emission. The equipment and the protocol are described hereafter.

### 2.1 Sound source

The sound source is generated by an underwater loudspeaker (Lubell Labs, LL 916H) controlled by an electronic device designed by the RTSys Company. The loudspeaker has a frequency response of +/- 10 dB between 0.5 kHz and 21 kHz, enabling the generation of sounds in this spectrum. The generated sound is determined by a theoretical signal (i.e. a wave file) and reproduced with a bias linked to the transfer function of the loudspeaker. The theoretical signal, chosen for this study, is a logarithmic chirp varying from 0.2 kHz to 50 kHz in 1 second, a bit larger than the theoretical frequency response of the loudspeaker. This signal is continuously emitted by the loudspeaker in an endless loop. In a preliminary study, several tests have been conducted in Lake Bourget (France) to characterize the response of the system.

To measure the generated sound at different angles from the speaker, 4 hydrophones (HTI 96) were placed at a fixed distance of 0.7 m from the sound source (Figure 1a). HTI96 hydrophones have a flat frequency response between 2 Hz and 30 kHz (+/- 2dB), enabling absolute measurement of the acoustic power in this frequency range. The entire system was deployed in a lake with an aluminum structure to ensure the relative position of the sensors (Figure 1b). To minimize the effect of this structure, all the sensors were attached to the structure with free ropes of 10 cm length. Several measurements of the emitted sounds were made by varying the depth of the system from 0.5 to 3.5 m and by changing the orientation of the loudspeaker (horizontal or vertical). The Power Spectral Density (PSD) of each emitted chirp monitored by the 4 hydrophones has been computed and plotted all together (Figure 2). It can be observed that the generated sounds have a spectral power between $10^{12}$ and $10^{14}$ $\mu Pa^2/Hz$ but do not have a flat frequency response due to the transfer function of the system. Overall, we observed that the monitored PSD was variable between the different tests that were conducted. The monitored power varied between +/- 3 dB between the quartiles 25 and 75, and between +/- 10 dB between the minimum and the maximum. The most important parameter influencing the emitted sounds was the directivity of the loudspeaker (horizontal or vertical positions). The emitted signals also did not vary when repeating the same signal in a fixed configuration of emission.

This preliminary study indicated that we would not be able to precisely predict the power emitted by the sound source during our experiments. The loudspeaker is deployed with a weighted rope from a bridge so that its orientation is uncertain when deployed on the river-bed. We therefore have an uncertainty concerning the initial power of the sound source ($P_{@1m}$) defined in the equation 1. This parameter will therefore be estimated for each experiment.

### 2.2 Hydrophone measurements at varying distances

Acoustic measurements were performed with HTI-96 hydrophones plugged to a EA-SDA14 recorder (RTSys company). Acoustic signals were stored in wav files at a sampling frequency of 156 kHz. The acoustic recorder and the hydrophone are shared by a Carlson river board, drifting during the measurements (Figure 3). Lagrangian measurements were preferred to fix-position measurements to optimize the signal to noise ratio. By measuring when drifting, the noises generated by the resistance of the river board against the flow are drastically reduced. The hydrophone was located under the river board at a constant depth from the water surface. The underwater loudspeaker is deployed at a fixed position on the river bed and emits a logarithmic chirp with an infinite loop of 1 second. During this time, several drift trajectories were made with the river board along the cross-section. As a first step, acoustic measurements were positioned using a synchronized GPS. This GPS equipment was damaged during the first field experiments requiring another way to position the hydrophone during the drifts. The cross-sectional distance of the hydrophone was monitored at start positions and considered as constant during the drift (i.e. drifts are considered parallel to the river banks). Secondly, longitudinal positions of the hydrophone during the drift were computed knowing the start position and by assuming a constant velocity of the river board:

$$x(t) = x_0$$
$$y(t) = y_0 + v_{drift}\, t \qquad\qquad (6)$$

Where $x$ and $y$ are respectively the cross-sectional and longitudinal positions of the hydrophone (m); $x_0$ and $y_0$ are the initial positions of the hydrophone monitored at the beginning of the drift; $v_{drift}$ is the mean velocity of the river board during the drift (m/s), computed as the travelled distance divided by the duration of the drift. The assumptions of parallel drifts at a constant velocity was supported by the fact that our field sites are straight reaches.

Finally, the position of the hydrophone is known over the time. The next section describes how the hydrophone signals were processed.

### 2.3 Signal Processing of the monitored acoustic waves

The use of a matched filter was chosen to detect the chirps in the hydrophone signals. When a chirp is detected, the position of the measurement is computed by matching the time of detection with the position of the hydrophone. Finally, knowing the position of the loudspeaker, the distance $r$, between the sound source and the measurement, is computed.

For each located chirp, a short-term spectrogram is computed using Hamming windows of $2^{12}$ points with 50% overlapping (Figure 4). Based on this spectrogram, several *PSDs* are computed. First, the PSD of the studied chirp (noted *$PSD_r$*) is computed by using the signal contained inside the black lines. The black lines correspond to the upper and lower limits of the octave band centered around the instantaneous frequency of the chirp. Secondly, the 95 percentile of the monitored power is computed (Merchant et al., 2013) in each frequency band. This PSD is used to represent the power of the ambient noise (*$PSD_{95}$*). In this example (Figure 4), one can particularly observe the harmonics generated by the loudspeaker when reproducing the theoretical logarithmic chirp. The ambient noise depends on the sounds that are naturally generated in the river (e.g. bedload impacts). To

ensure that the chirp is not affected by ambient noise, we decided to keep only the chirps that are at least twice more powerful than the ambient noise (i.e. $PSD_r > 2\ PSD_{95}$).

At this point, we can propose a protocol to monitor the $PSD$ of an emitted chirp at varying distances from its point of emission.

### 2.4 Fitting propagation laws

The acoustic power of each chirp measured at a distance $r$ was computed by integrating $PSD_r$ in third-octave bands. For the $j^{th}$ third-Octave band, $P_{i,j}$ is the acoustic power of the $i^{th}$ measurement made at a distance $r_i$ from the loudspeaker. Using the theoretical model (eq.4) and assuming one model of geometric spreading loss (cylindric or spherical), the estimated acoustic power $\widetilde{P_{i,j}}$ in function of $r_i$ is:

$$\widetilde{P_{i,J}} = P_{@1m,j}\ G(r_i)\ e^{-2\alpha_j r_i} \tag{7}$$

where $P_{@1m,j}$ and $\alpha_j$ are parameters to fit for each third Octave band $j$. These parameters were estimated with a non-linear least square algorithm on the log values of power. It means that $P_{@1m,j}$ and $\alpha_j$ were estimated by minimizing the following term: $\sum_{i=1}^{N}\left[\log(\widetilde{P_{i,J}}) - \log(P_{i,j})\right]^2$ where $N$ is the total number of observed chirps.

For each frequency band, the fit was characterized by a coefficient of correlation between the log values of the estimated power ($\widetilde{P_{i,J}}$) and the log values of the measured power ($P_{i,j}$). Finally, the residuals (dB) of the fits were computed using the following relationship: $\frac{1}{N}\sum_{i=1}^{N}\left|10\log(\frac{\widetilde{P_{i,J}}}{P_{i,j}})\right|$. The residuals are the average variation of the data set around the fitted law, it represents the dispersion of the data set.

In summary, the source power ($P_{@1m}$) and the attenuation coefficient ($\alpha$) were estimated by fitting a propagation law (equation 4) to power measurements made at several distances from the loudspeaker. Estimations were made by considering third-Octave bands, therefore enabling the estimation of $P_{@1m}$ and $\alpha$ in several frequency bands. Note that these estimations were done by considering either a cylindrical (eq. 2) or spherical model (eq. 3).

### 2.5 Field sites

The protocol presented in the previous section was applied in 7 field sites located in the French Alps. Their characteristics are presented in the Table 1. The mean bed-slope of the studied reaches varies from 0.05 to 1 %, and the width of the cross-section from 8 to 60 m. The roughness (or the surface particle-size distribution) of the river bed is a difficult parameter to measure, particularly in rivers that are not wadable. This aspect of bed roughness was approached by doing Wolman measurements on the closest emerged bars. The surface $D_{84}$ of emerged bars varies from 20 to 150 mm. Hydraulic parameters (discharge, surface velocity and mean water depth) were obtained by using several methods (acoustic Doppler current profiler, SVR Radar Gun or existing gauging station) depending on the field sites. Finally, the measurement of suspended sediment load was achieved with a turbidimeter (Visoturb, WTW).

## 3 Results

### 3.1 The Leysse River

Data from the Leysse River are presented in the Figure 5 (see also Appendix A). It represents the acoustic power received by the hydrophone at different distances from the underwater loudspeaker. As an example, the results obtained with the third-octave band centered on 1 kHz are shown in Figure 5, this data set has been obtained with 27 drifts of the river board. The effect of source location has been tested by varying the source location in the river cross-section. It has been found that the result was insensitive to source location in this river. Spherical and cylindrical models of propagation losses have been fitted with a least square procedure on the logarithmic values of the acoustic power. Two parameters are obtained, the initial power of the sound source ($P_{@1m}$) and an attenuation coefficient ($\alpha$). This procedure was repeated on each third-octave band to obtain the variation of these parameters with frequency (Appendix A).

Results of the fits are shown in the Figure 6 for the Leysse river experiment. Logically, attenuation coefficients that are estimated with cylindrical spreading loss exhibit higher values than coefficients estimated with spherical spreading loss. However, they behave similarly with frequency variations. At low frequency, approximatively below $10^3$ Hz, attenuation coefficient is higher. This result was expected because of the existence of a cutoff frequency (Geay et al., 2017b). The cutoff frequency is dependent on the water depth (mean water depth of 0.95 m), the sound speed in water (assumed to be equal to 1500 m/s) and the sound speed in the sediment layer. Typical values of sound speed in sea floor materials (from silt to gravel) were observed to vary between 1550 to 2000 m/s (Jensen et al., 2011), depending on many factors such as the type of materials, grain-sizes or porosity (Hamilton and Bachman, 1982). Using sound speed of 1550 and 2000 m/s in the sediment leads to cutoff frequencies of 1500 Hz and 600 Hz, respectively, which is consistent with our observation.

Above $10^3$ Hz, attenuation coefficient increases with frequency: acoustic waves are more attenuated at higher frequencies. Considering the estimation of the sound source power, it is observed that the cylindrical model best reproduces the power monitored in the experiment made in the Bourget lake (the median value is represented in the Figure 6b). Using a spherical model, we overestimate the power of the sound source by approximatively one order of magnitude. However, as we will see for other experiments, the best estimation of the sound source power is sometimes obtained with spherical spreading loss model.

In the Figure 6c, the residuals of the regression represent the dispersion of the data around the fit. It has been computed as the mean square difference between data and fits. In the Leysse river, we observed that the power of the reception fluctuates between 2 and 3 dB around the fits.

Finally, considering the correlation coefficients of the fitted laws (Figure 6d), we cannot make a distinction between spherical or cylindrical spreading loss models.

### 3.2 Propagation laws in several rivers

Propagation properties of several rivers were investigated. For some of the rivers, experiments were done at different hydrodynamic conditions (Table 1). For the discharge investigated, hydrodynamic conditions were not enough variable to observe major differences in the results. We therefore decided to gather data to propose a unique result for each river. The

data set is presented in the Appendix A, for all rivers and for different frequency bands. A first result concerns the estimated power of the sound source ($P_{@1m}$) emitted during the experiments (Figure 7). Compared with the measurements made in the Bourget lake, it can be observed that the estimation of the sound source power is overestimated when using a spherical model and underestimated when using a cylindrical model of the geometric spreading loss. Considering the correlation coefficients of the data to the fits, we did not observe a significant difference between the models. Based on these observations, we are not

able to argue that geometric spreading is cylindrical or spherical in these rivers. In the following, all the results are presented by assuming a cylindrical, spreading-loss model.

The attenuation coefficients obtained for each river are presented as a function of frequency (Figure 8). From the Isère to the Arve river, we can observe that the attenuation coefficient varies by more than one order of magnitude (Figure 8b). Looking at the linear representation (Figure 8a), we see that the variation of the attenuation coefficient with frequency is different from

case to case. It increases faster for rivers having the largest attenuation coefficients. Note that minimal and maximum frequencies of the observations are variable from one river to another. At low frequency, observations are limited by the cutoff frequency which is inversely proportional to the water depth (Geay et al., 2017b). At high frequencies, measurements are limited by too strong attenuation of the emitted acoustic waves.

The Table 2 contains, for each river, a summary of the results obtained by fitting a cylindrical propagation model to the data.

All the parameters indicated in this table are an average of the values obtained between 1 and 10 kHz. It can be observed that the correlation coefficients vary from 0.4 to 0.8. We observed that the lowest correlation coefficients were obtained for the largest rivers (Isère and Romanche rivers with section width of 60 and 33 m, respectively) and may be representative of cross-sectional variations that have not been considered in this study. The residuals vary from 2 to 6 dB. Rivers having largest attenuation coefficients seem to have larger residuals: the dispersion of the monitored acoustic power is larger when the

attenuation is larger. Finally, the maximum distance of the monitored chirps represents the maximum distance from the hydrophone to the underwater speaker where we were able to record the chirps with a sufficient signal to noise ratio. The smaller the attenuation coefficient, the larger the maximum distance of the observation. Note that the maximum distance is also dependent on operational issues.

## 4 Discussion

### 4.1 Attenuation processes in rivers

During our field campaign, it has been found that attenuation coefficients were variable from one river to another. The attenuation due to freshwater vary from $10^{-9}$ to $10^{-3}$ nepers/m from 1 to 100 kHz (Fisher and Simmons, 1977). The attenuation due to water only do not explain the coefficients of attenuation that were found in this study. In this section, we wonder how propagation properties are related to typical characteristics of the rivers (e.g. slope, water depth). As shown in Figure 8 the dependency of the attenuation coefficient to frequency do not follow a simple law.

At low frequency, around 1 kHz, acoustic wave propagation should be affected by wave guide properties. The river could be considered as an acoustic wave guide where sounds are partly trapped between the water surface and the river bed (Geay et al., 2017b): this problem is known as the Pekeris waveguide. Theoretically, in a perfect medium without attenuation, it can be shown that acoustic waves having frequencies lower than the cutoff frequency are exponentially decaying with horizontal distance (Jensen et al., 2011). The cutoff frequency $f_{cutoff}$ (Hz) is dependent on the wave guide characteristics, water depth and sediment layer acoustic properties, as shown in the following equation:

$$f_{cutoff} = \frac{c_s c_w}{4h\sqrt{c_s^2 - c_w^2}} \tag{8}$$

Where $h$ is the water depth (m), $c_s$ and $c_w$ are sound celerity (m/s) in the sediment layer and in water, respectively. Cutoff frequencies have been estimated in each river, by assuming a fixed sound speed of 1600 m/s in the sediment layer and using the mean water depth monitored (Figure 8b). Estimated cutoff frequencies are approximatively located around the minimum of the observed attenuation coefficient. Our ability to precisely determine a cutoff frequency is limited. First, the acoustical properties of river beds are unknown, depending on lithology, grain sizes, porosity and heterogeneity of the materials constituting the river bed. Secondly, the water depth is not constant over the investigated sections but varies from the banks to the middle of the river. For these reasons, cutoff frequencies are rough estimates and do not perfectly correspond to the observed local minimum of attenuation coefficient. Note also that different hydrodynamic conditions were investigated for some rivers. Varying water depth results in different cutoff frequencies and relative positions of the hydrophone between water surface and streambed. These two parameters have been observed to modify the response of the hydrophone (Geay et al., 2017b) in the lower frequency range, around the cutoff frequency. The range of hydrodynamic conditions that was investigated in this study did not enable the observation of such effects.

The variation of attenuation coefficients at higher frequencies is here discussed. As attenuation properties are frequency dependent, it is common to characterize the attenuation in mediums by giving a value of the attenuation coefficient per wavelength (eq. 5). Attenuation coefficients per wavelength (nepers) are presented in the Figure 9 for frequencies higher than the local minimum of $\alpha$ (nepers/m). Except for the Isère river, we can observe that $\alpha_\lambda$ is almost constant with frequency, which in turns means that $\alpha$ (nepers/m) varies almost linearly with frequency. Note that the maximum frequency analyzed in the Figure 9 is varying from one river to another and shows some inverse correlation with the average $\alpha_\lambda$. In rivers with high

attenuation (high $\alpha_\lambda$), the recording of the emitted chirps was not possible in the highest frequency range, due to a too fast decrease of chirp power with distance. This observation reveals that the measurement of bedload sounds generated by the smallest particles (i.e. highest frequencies generated) should be local, or even impossible in rivers having too high attenuation coefficients. In the following, each river is characterized by an average value of $\alpha_\lambda$ (Table 3) and is compared to river characteristics (Table 1; Figure 10). Looking at the relationship between $\alpha_\lambda$ and the slope measured at the local reach (i.e. 100 meters downstream and upstream from the bridge where experiments were undertaken), we can observe that there is good correlation: higher attenuation coefficients were obtained for steeper rivers. As for slope, surficial granulometry of the emerged bars (**$D_{84}$**) are also well corelated to $\alpha_\lambda$: larger roughness (i.e. larger **$D_{84}$**) induces larger attenuation of the acoustic waves. Surface velocity or water depth seems to be less robust explanatory variables of $\alpha_\lambda$. The possible influence of typical nondimensional numbers has also been tested. The ratio of the water depth over the **$D_{84}$** and the Froude number were used by Tonolla et al. (Tonolla et al., 2009, 2010). They found that they were the main hydrogeomorphological variables explaining the differences in passive acoustic signals in field experiments. Small ratio of the relative submergence (i.e. small **$h/D_{84}$**) induce breaking waves or water plunging directly in the water column, entraining bubbles in the water column. These hydraulic mechanisms are sources of noise generated by oscillating air bubble in the water column as it is observed for breaking waves in marine environment (Deane, 1997; Norton and Novarini, 2001). In our study, entrained air bubbles could explain the increase of attenuation coefficient in rivers having rough beds. It is indeed known that the presence of air bubbles increases the attenuation of acoustic waves (Deane, 1997; Norton and Novarini, 2001) because of the heterogeneity of the medium constituted of water and air which have very different acoustic impedances. Also, as observed in a flume experiment (Wren et al., 2015), the bed roughness itself is a source of attenuation, larger roughness involving higher attenuation. Finally, both processes, rough boundaries and entrained air bubbles could explain our observations by causing concomitantly higher attenuation of the acoustic wave. The river bed roughness should be the best characteristic enabling the prediction of acoustic wave propagation properties in river. However, this parameter is not easy to measure. It is sometimes difficult to access the riverbed, and surface grain size distributions are known to be variable in space. The local slope of the reaches is easier to measure and, even if less meaningful, should be a more robust parameter to infer propagation properties of a river.

## 4.2 Recommendation for monitoring bedload with hydrophones

This study was done to improve our ability to better use the measurements of bedload self-generated noise in rivers. This section aims at giving an example on the use of attenuation coefficients in a simple case. Let us consider an infinite river bed with a homogeneous repartition of sound sources over the river bed. Bedload impacts generate a constant spectral power per surface unit noted **$PSD_s$** ($\mu$Pa$^2$/Hz/m$^2$). If sound sources are random and independent noise sources (Thorne, 2014), the acoustic power measured by a hydrophone can be written as a sum of the power of all sound sources:

$$PSD_h(f) = \int_d^\infty \frac{2PSD_s(f)}{rh} e^{-2\alpha(f)r} 2\pi r dr \tag{9}$$

Where $PSD_h$ is the spectral power monitored by a hydrophone in a fixed position ($\mu Pa^2$/Hz), $h$ is the water depth (m), $d$ the distance of the hydrophone above the river bed (m) and $r$ the horizontal distance from the hydrophone (m). From equation (9), it follows:

$$PSD_h(f) = \frac{2\pi PSD_s(f)}{h\alpha(f)} e^{-2\alpha(f)d} \tag{10}$$

Considering that $0< \alpha<<1$, it follows that $PSD_h$ is inversely proportional to the attenuation coefficient as the exponential term tends to 1. This has several implications for the use of bedload monitoring using passive acoustics. First, as the attenuation coefficient could be variable from one reach to another, the acoustic power of bedload SGN could be variable from one reach

to another even if bedload fluxes are similar. Secondly, as observed in Figure 8, attenuation coefficients are variable with frequency. It means that the frequency content of bedload SGN spectra is modified by propagation effects, which in turns means that the shape of monitored spectra are not only related to grain size distributions (Petrut et al., 2018; Thorne, 2014) but also to propagation properties. Therefore, in order to estimate grain size distribution, measured spectra should be corrected for propagation effects before any inversion procedure. From equation 10 ($\alpha<<1$), a better estimate of the sound generated by

bedload transport could be done by multiplying the monitored sound pressure levels by the attenuation coefficient ($\alpha > 0$):

$$PSD_s(f) = \frac{h}{2\pi}\alpha(f)PSD_h(f) \tag{11}$$

The power generated by bedload sounds is proportional to the power of measured sounds multiplied by the attenuation coefficient. This simple operation enables us to get an unbiased measurement of the sound generated by bedload impacts, and

20 therefore a more robust proxy for bedload transport monitoring in rivers. To achieve the estimation of sounds that are generated by bedload transport ($PSD_s$), both measurements of propagation properties ($\alpha$) and ambient sounds ($PSD_h$) are needed. Note that equation 11 was obtained by assuming sound sources (i.e. bedload fluxes) that are homogeneously distributed. As this hypothesis will rarely be valid, more realistic inverse methods should be invented to estimate the real sounds ($PSD_s$) generated by bedload transport and its spatial distribution.

**5 Conclusion**

A simple model for acoustic wave propagation in rivers has been investigated in this study. It considers that the power of acoustic waves decreases with distance by spreading effects (cylindrical or spherical models) and with an additional

exponential term including other propagation effects (e.g. volume attenuation, scatter by rough boundaries). The model was used to interpret the attenuation properties of a controlled sound source in several rivers having different hydrogeomorphic characteristics. Our tests were not able to distinguish whether spherical or cylindrical models should be used, both models being valid. The exponential attenuation coefficient ($\alpha$ in nepers/m) has been found to vary with frequency and with the type of river considered. Two types of attenuation regimes have been observed. Below the cutoff frequency, which is inversely proportional to the average water depth, $\alpha$ decreases with increasing frequency until a local minimum is reached. Reaches with large water depth should therefore be selected for doing passive acoustic measurements. The cutoff frequency should be sufficiently low to listen to the coarsest grains of bedload transport. Above the local minimum (i.e. the cutoff frequency), attenuation coefficients increase almost linearly with frequency. The higher frequency regime has been characterized by a constant attenuation coefficient per wavelength ($\alpha_\lambda$ in nepers). It has been found that $\alpha_\lambda$ was well correlated to the slope of the river-bed reaches (and to the surface $D_{84}$ of the emerged bars as well), where $\alpha_\lambda$ is higher for higher bed slopes of the river. Assuming that river-bed slope and surface $D_{84}$ of bars are good proxies for the river-bed texture, it can be concluded that attenuation properties is dominated by processes related to the river-bed roughness at high frequencies, including the entrainment of air bubbles in the water column and scattering effects on rough boundaries. As shown in the discussion, the acoustic power monitored by a hydrophone, in a fixed position, is almost inversely proportional to the attenuation coefficient at a given frequency. As consequences, the spectra of bedload SGN that are measured in rivers are modified by the variations of attenuation coefficients with frequency. As attenuation is higher at high frequencies, acoustic signals that are monitored by a hydrophone are shifted to lower frequencies compared to the sound really generated by bedload impacts. As shown for an idealized case with an infinite riverbed and homogeneous bedload sound sources, the real sounds generated by bedload can be estimated by correcting the hydrophone signal by the propagation laws of acoustic waves in rivers.

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

**Tables**

**Table 1 : Field site caracteristics**

| River | Local slope (%) | Width of the cross-section (m) | GSD of emerged bars $[D_{50}-D_{84}]$ (mm) | Date of field experiments | Water discharge $(m^3/s)$ | mean water depth (m) | mean surface velocity (m/s) | suspended sediment concentration (g/L) |
|---|---|---|---|---|---|---|---|---|
| Arve | 0.75 | 14 | [70-120] | 2017/06/27 | 38 | 1.25 | 2.3 | 0.35 |
| | | | | 2017/06/29 | 29 | 1.1 | 1.95 | - |
| Grand-Buëch | 0.7 | 13 | [30-66] | 2017/04/12 | 5.5 | 0.35 | 1.5 | <0.05 |
| | | | | 2017/05/15 | 12.5 | 0.55 | 1.85 | <0.05 |
| Isère | 0.05 | 60 | [23.5-36.5] | 2017/03/08 | 171 | 2.4 | - | 0.1 |
| | | | | 2017/03/28 | 150 | 2.3 | - | 0.06 |
| | | | | 2017/06/06 | 237 | 2.8 | 1.85 | 0.6 |
| Leysse | 0.1 | 18 | [39-68] | 2017/03/09 | 17 | 0.95 | 1.2 | <0.05 |
| Romanche | 0.13 | 33 | [20-39] | 2017/06/14 | 55 | 1.2 | 1.85 | 0.14 |
| Sarenne | 0.13 | 8 | [4-8] | 2017/04/05 | 1.3 | 0.3 | 0.7 | <0.05 |
| Séveraisse | 1.0 | 12.5 | [32-75] | 2017/04/25 | 5 | 0.4 | 1.8 | <0.05 |

**Table 2 : Average results over frequency (1-10 kHz) of the parameters of the fit using cylindrical geometrical spreading.**

| River | $\alpha$ (nepers/ m) | Corr. coeff. of the fit ($r^2$) | Residuals (dB) | Maximum distance of the monitored chirps (m) |
|---|---|---|---|---|
| Arve | 0.26 | 0.5 | 6 | 11 |
| Grand-Buëch | 0.08 | 0.7 | 4 | 31 |
| Isère | 0.008 | 0.4 | 3 | 80 |
| Leysse | 0.01 | 0.8 | 2 | 39 |
| Romanche | 0.02 | 0.5 | 4 | 59 |
| Sarenne | 0.099 | 0.8 | 6 | 57 |
| Séveraisse | 0.215 | 0.8 | 5 | 22 |

**Table 3:** frequency bands where $\alpha_\lambda$ was observed to be almost constant with frequency and average values of $\alpha_\lambda$ in this frequency range.

| River | Frequency range [fmin-fmax] (kHz) | Average $\alpha_\lambda$ (nepers/wavelength) |
|---|---|---|
| Arve | [1-13] | 0.125 |
| Grand-Buëch | [1.6-20] | 0.032 |
| Isère | [1-40] | 0.003 |
| Leysse | [2.5-40] | 0.005 |
| Romanche | [2.5-40] | 0.004 |
| Sarenne | [8-40] | 0.005 |
| Séveraisse | [2.5-13] | 0.085 |

**Figures**

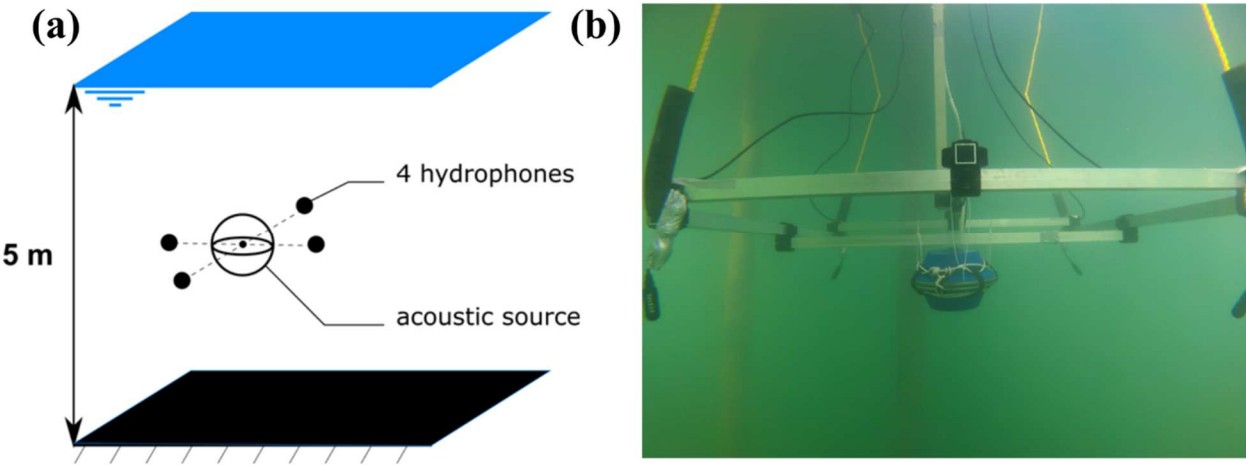

**Figure 1: (a) schematic design of the test characterizing the system of emission; (b) photography of the immerged system in the lake of the Bourget (France).**

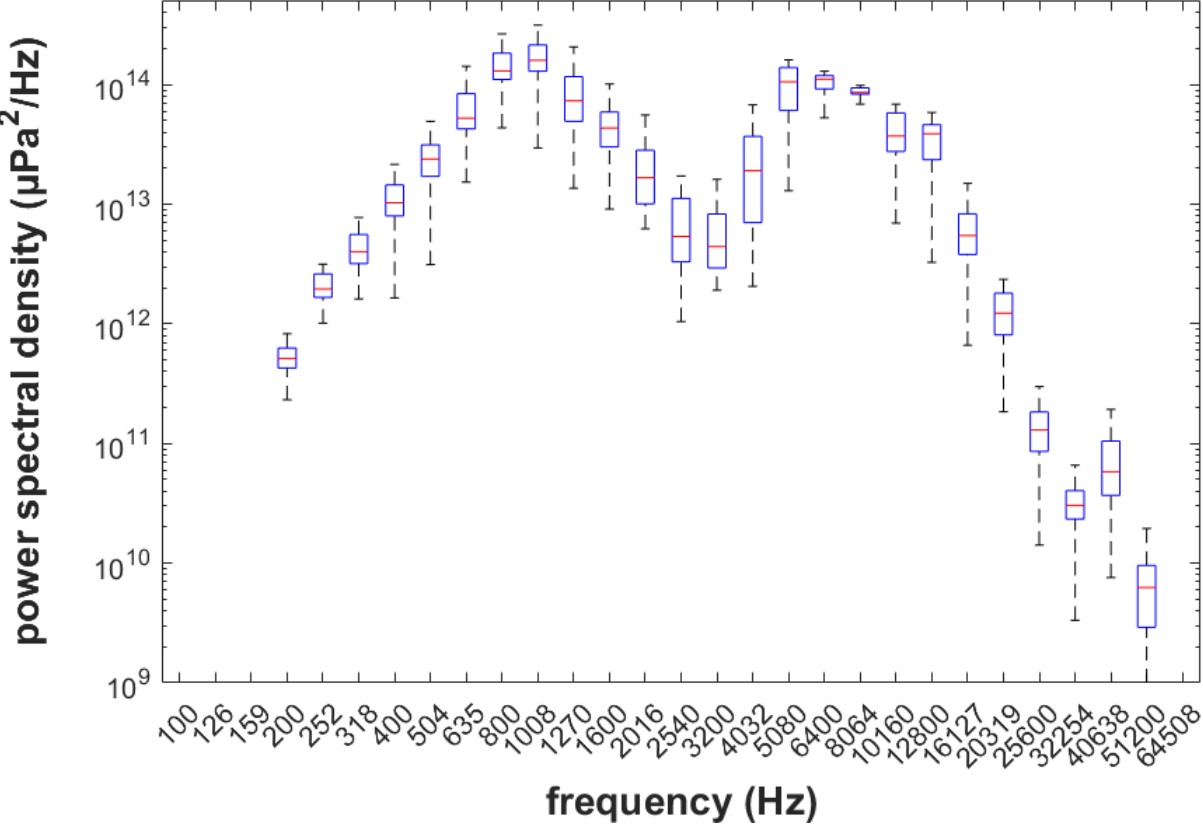

**Figure 2: Power Spectral Densities (µPa$^2$/Hz) of the logarithmic chirps emitted by the loudspeaker.**

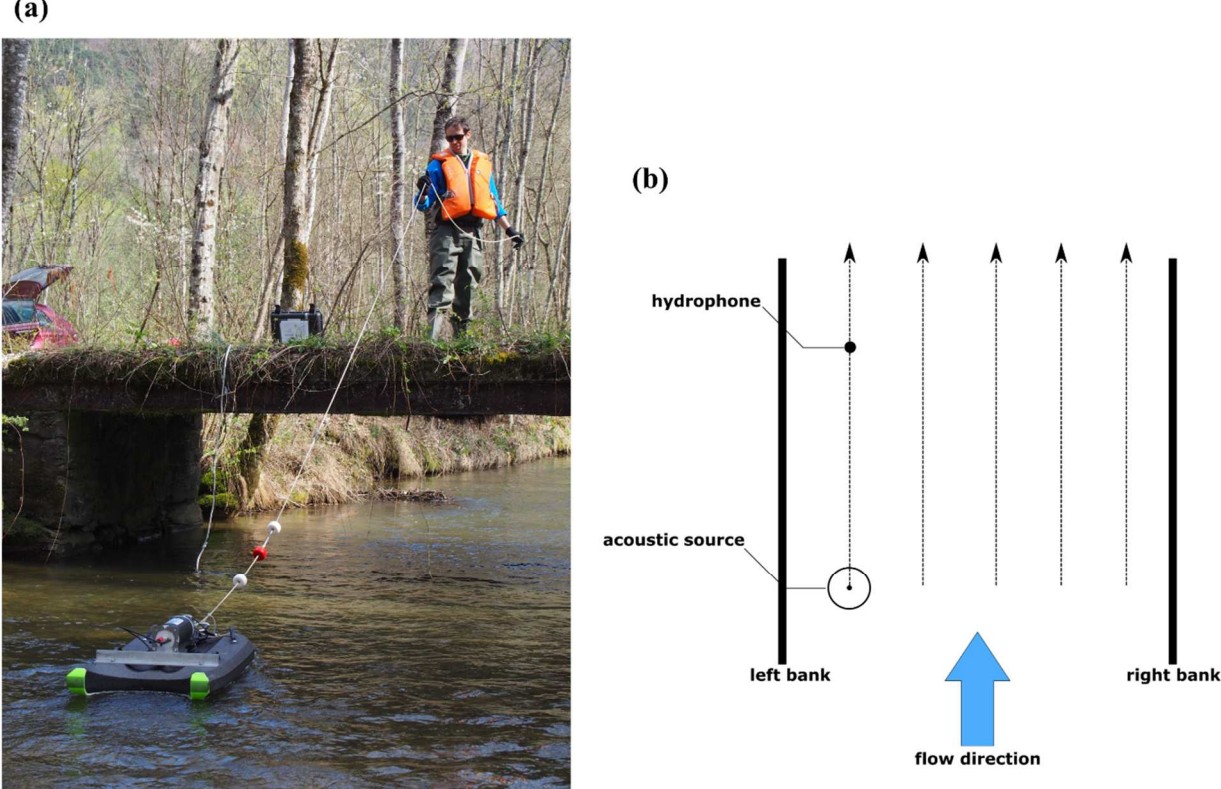

**Figure 3: (a) Drifting board sharing the hydrophone and the acoustic recorder; (b) Drift trajectories of the recorder during the measurements.**

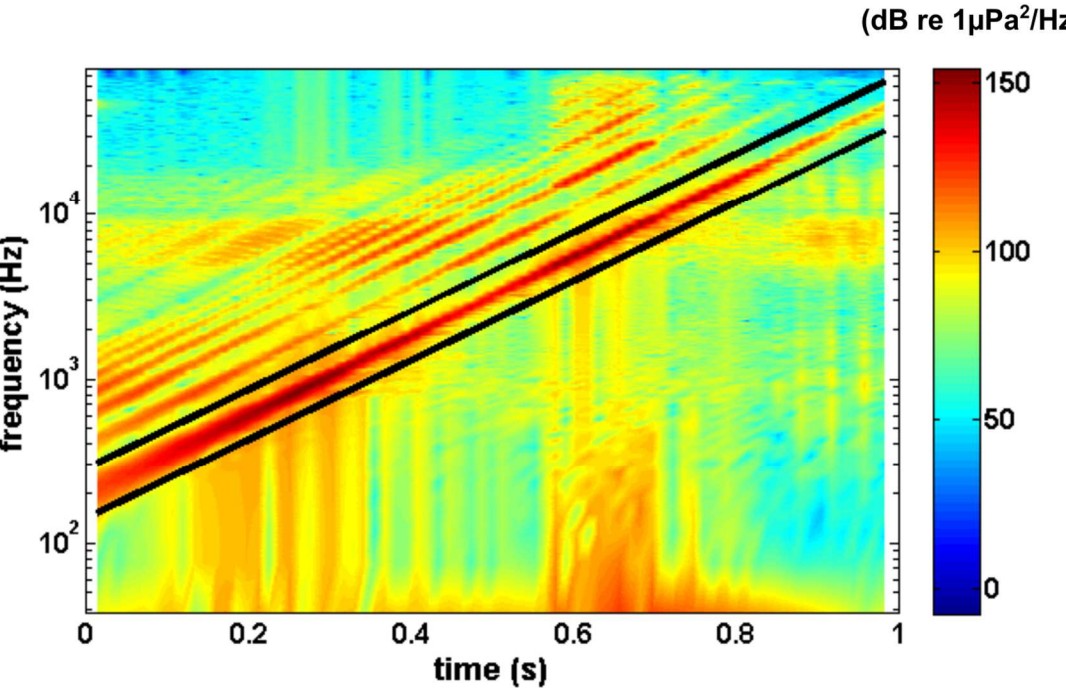

**Figure 4: Short-term spectrogram of a chirp monitored by a hydrophone in the Leysse River. The black lines indicate the octave band centered around the instantaneous frequency of the theoretical logarithmic chirp.**

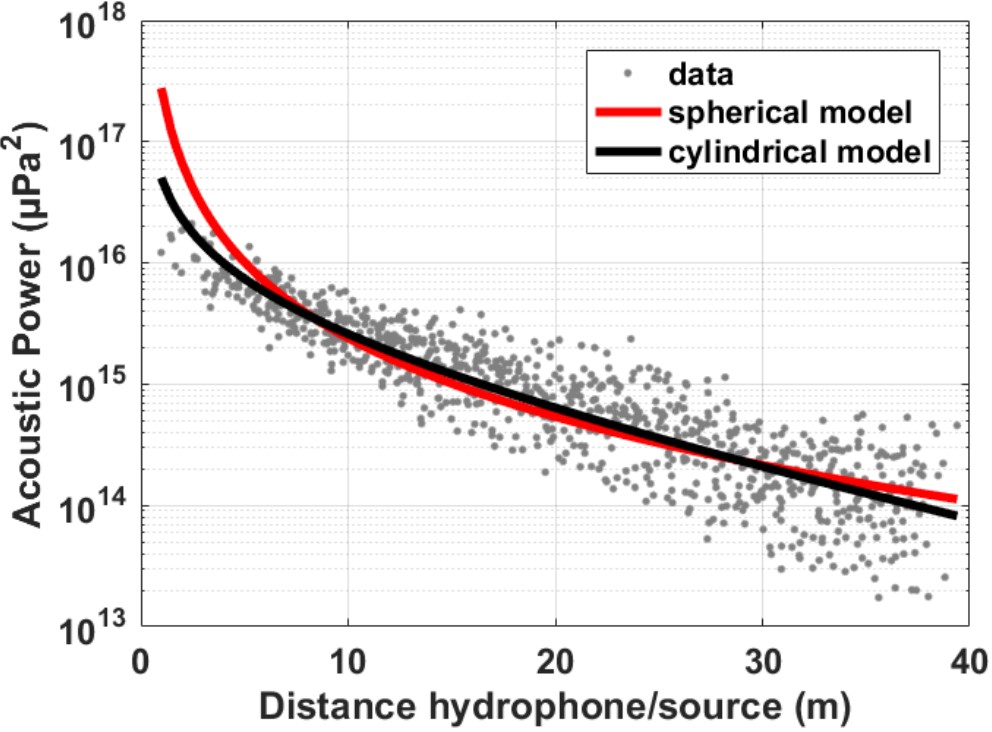

**Figure 5: Data from the Leysse river. Measured acoustic power (µPa$^2$) in function of the distance between the hydrophone and the active source. Results obtained in the third-Octave band centered on 1 kHz. Spherical and cylindrical fits are in thick lines. Fits and data sets are presented for other frequencies in the Appendix A.**

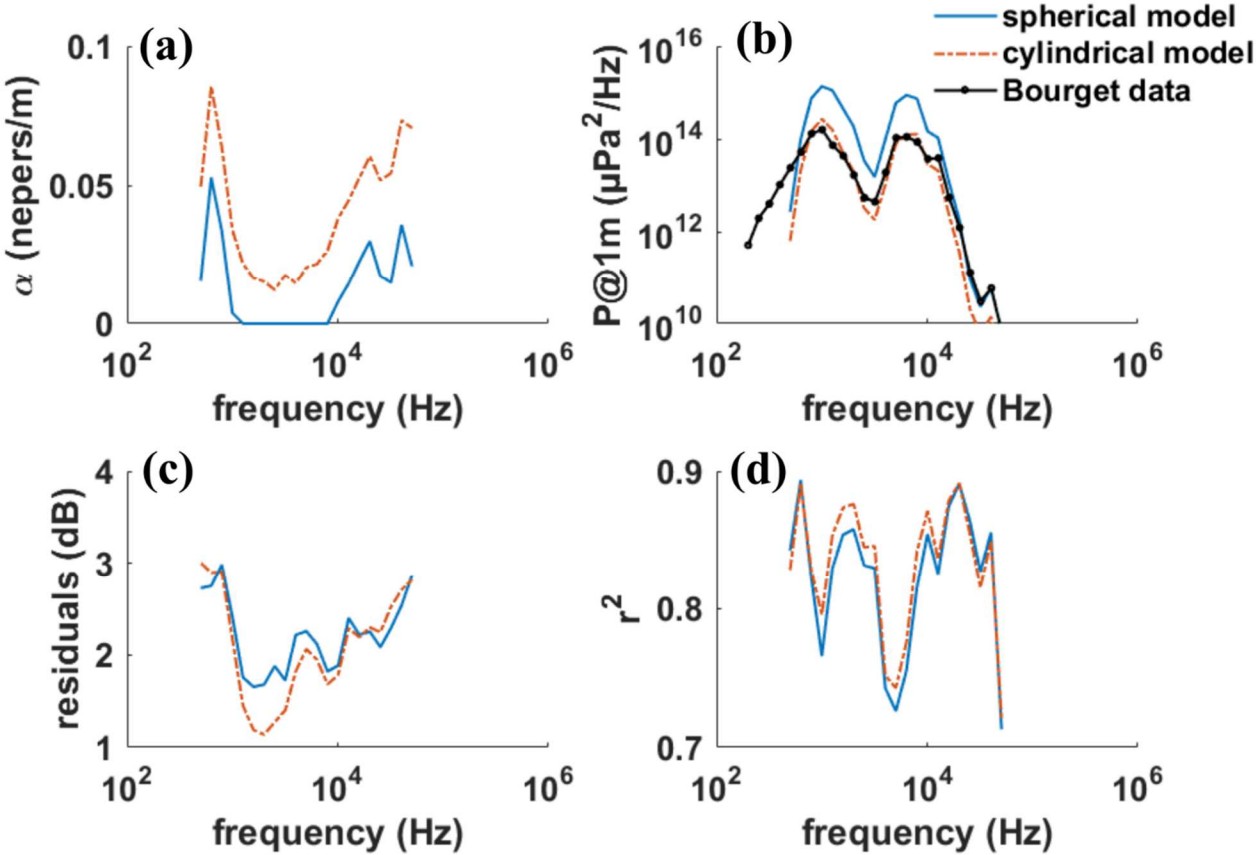

**Figure 6: Data obtained for the Leysse river (a) Attenuation coefficient (nepers/m) ; (b) Sound source power spectral density (P@1m in µPa²/Hz) estimated with spherical/cylindrical models. Data from the Bourget lake are the median values of the measurement presented in Figure 2; (c) Residuals of the regression (dB); (d) Squared correlation coefficient of the fits.**

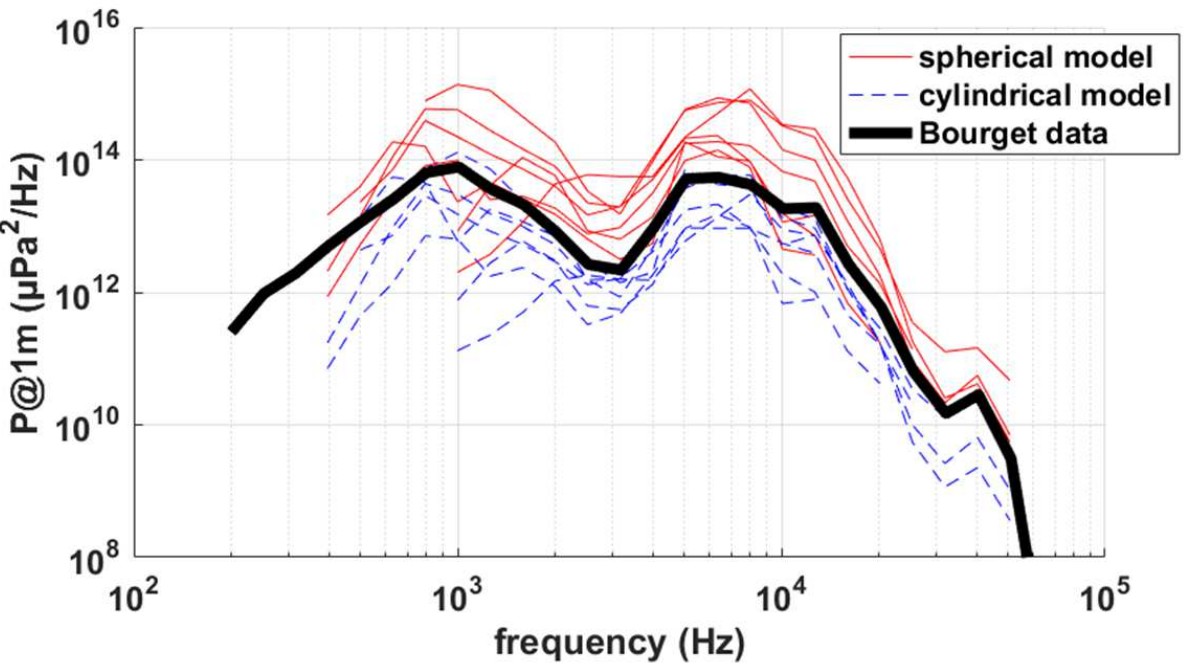

**Figure 7: Power spectral density of the source power (P$_{@1m}$ in μPa$^2$/Hz) estimated with spherical and cylindrical models for all experiments made in rivers and measured in the Bourget lake.**

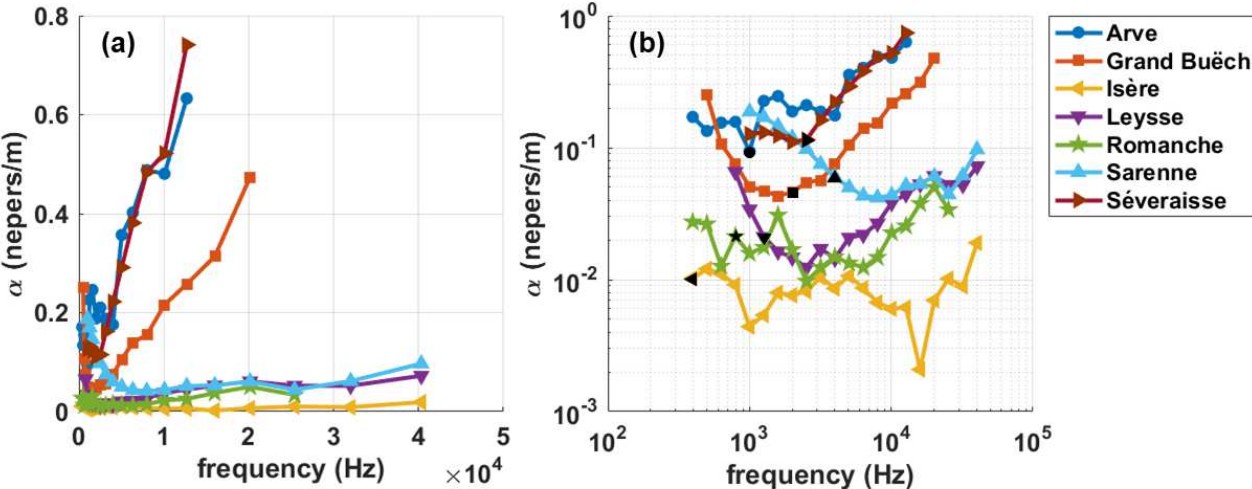

**Figure 8: attenuation coefficient (α in nepers/m) obtained when using a cylindrical model of the geometrical spreading loss: (a) linear and (b) logarithmic scales. Black symbols indicate the cutoff frequency computed with eq. (8), sound speeds of 1500 m/s and 1600 m/s in the water and sediment layer, respectively.**

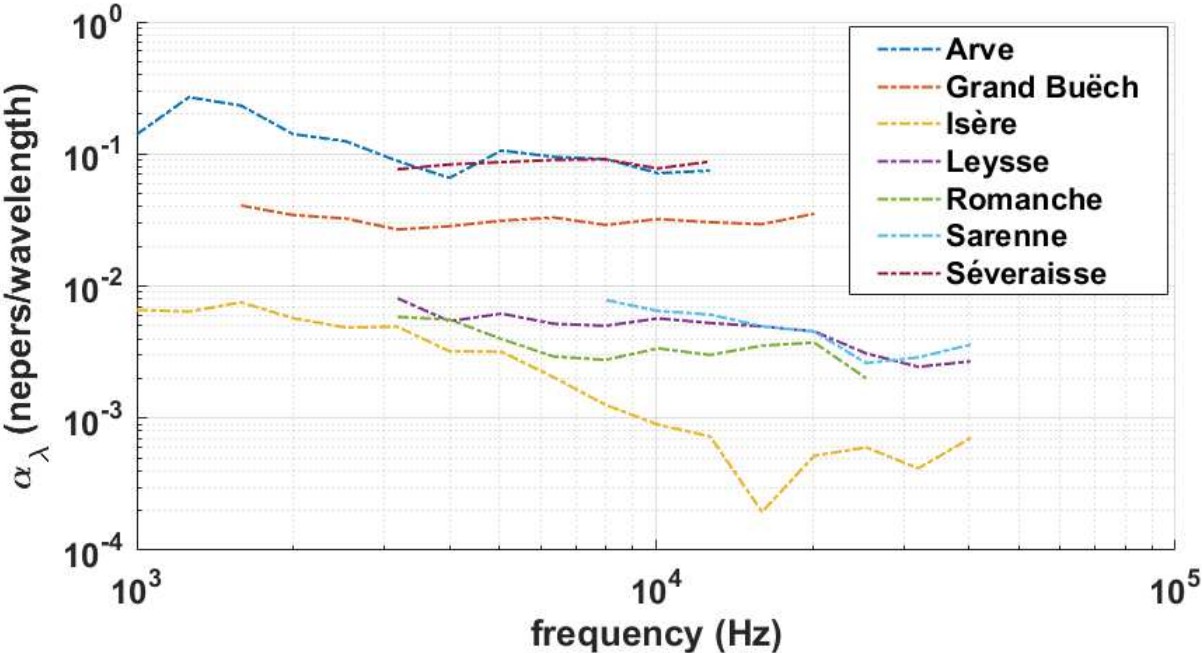

**Figure 9: attenuation coefficient per wavelength (nepers) in function of frequency (Hz) above the cutoff frequency.**

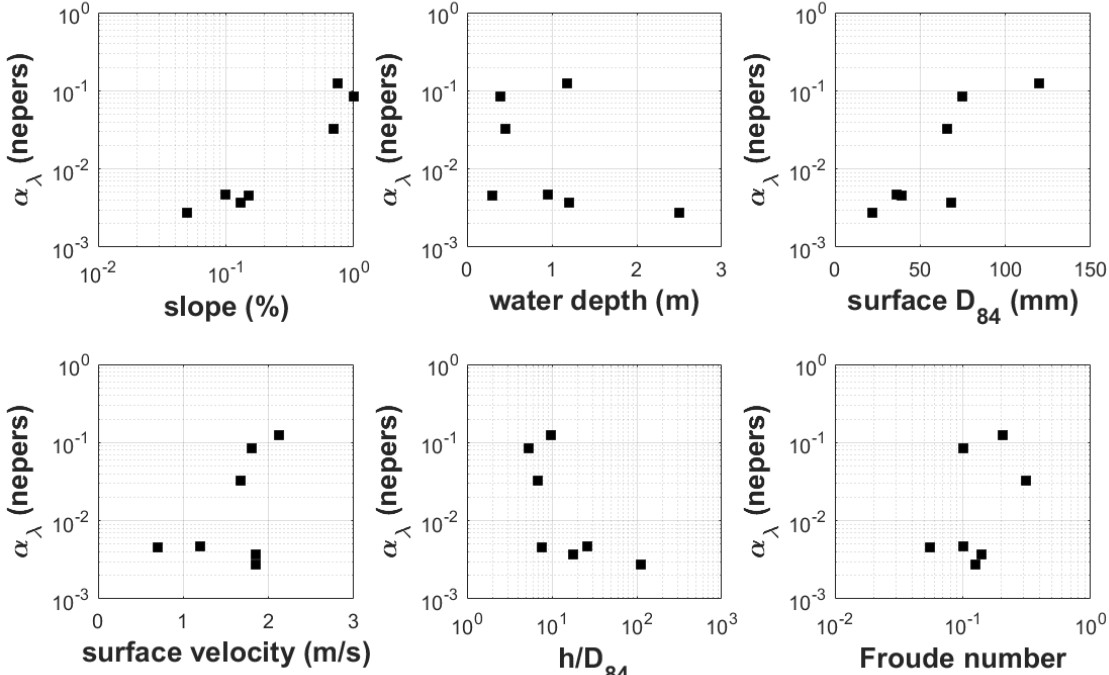

**Figure 10: representation of the attenuation coefficient per wavelength ($\alpha_\lambda$ in nepers) in function of river characteristics: (a) local slope (%); (b) average water depth (m); (c) surface $D_{84}$ (mm) of the closest emerged bars; (d) average surface velocity (m/s); (e) ratio of water depth (m) over surface $D_{84}$ (m); (f) Froude number computed with average flow velocity (m/s) and water depth (m).**

**Appendix A**

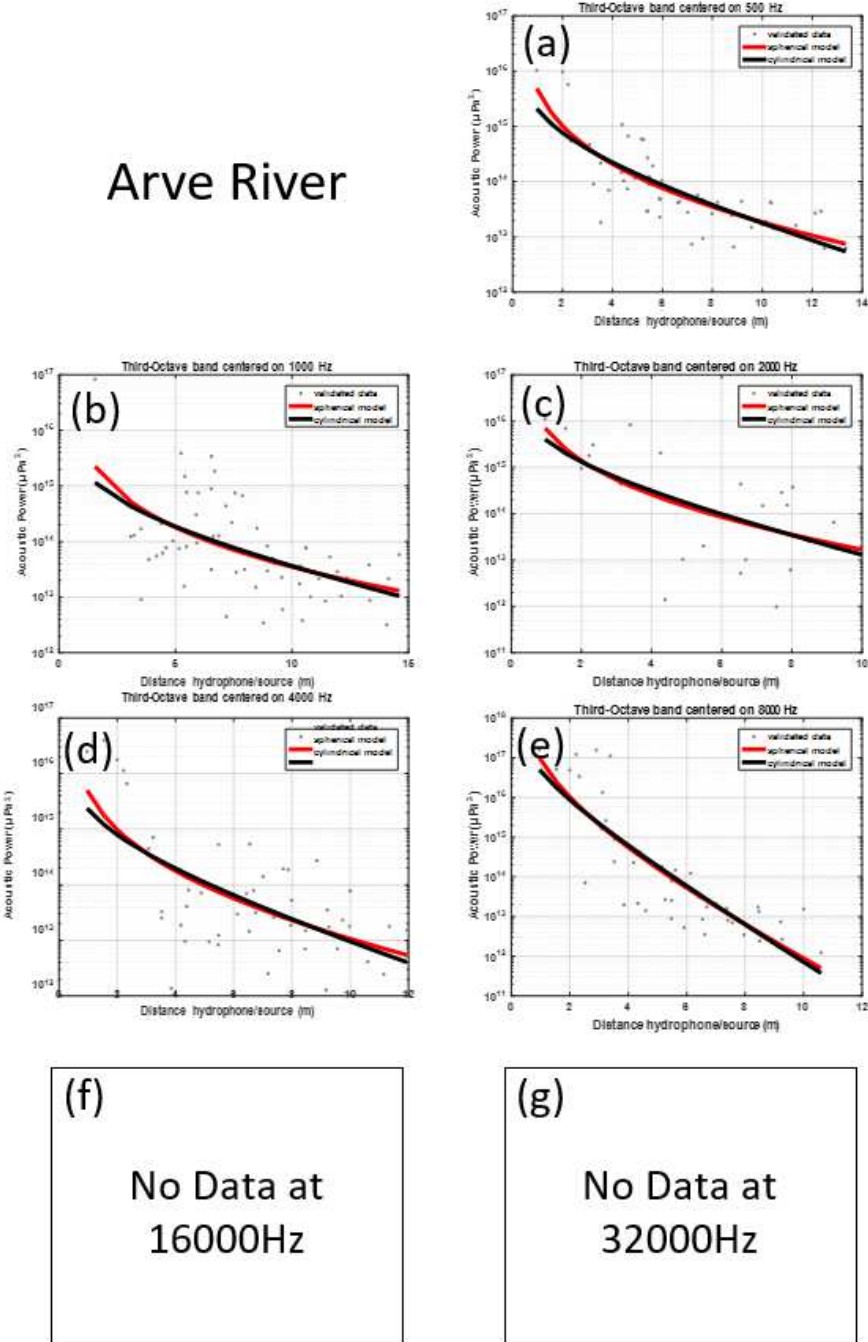

**Figure A1: Data obtained in the Arve River. Measured acoustic power (µPa2) in function of the distance between the hydrophone and the active source. Results obtained in third-Octave bands centered on (a) 500Hz, (b) 1000 Hz, (c) 2000 Hz, (d) 4000 Hz, (e) 8000 Hz, (f) 16000 Hz and (g)32000 Hz. Spherical and cylindrical fits are in thick lines.**

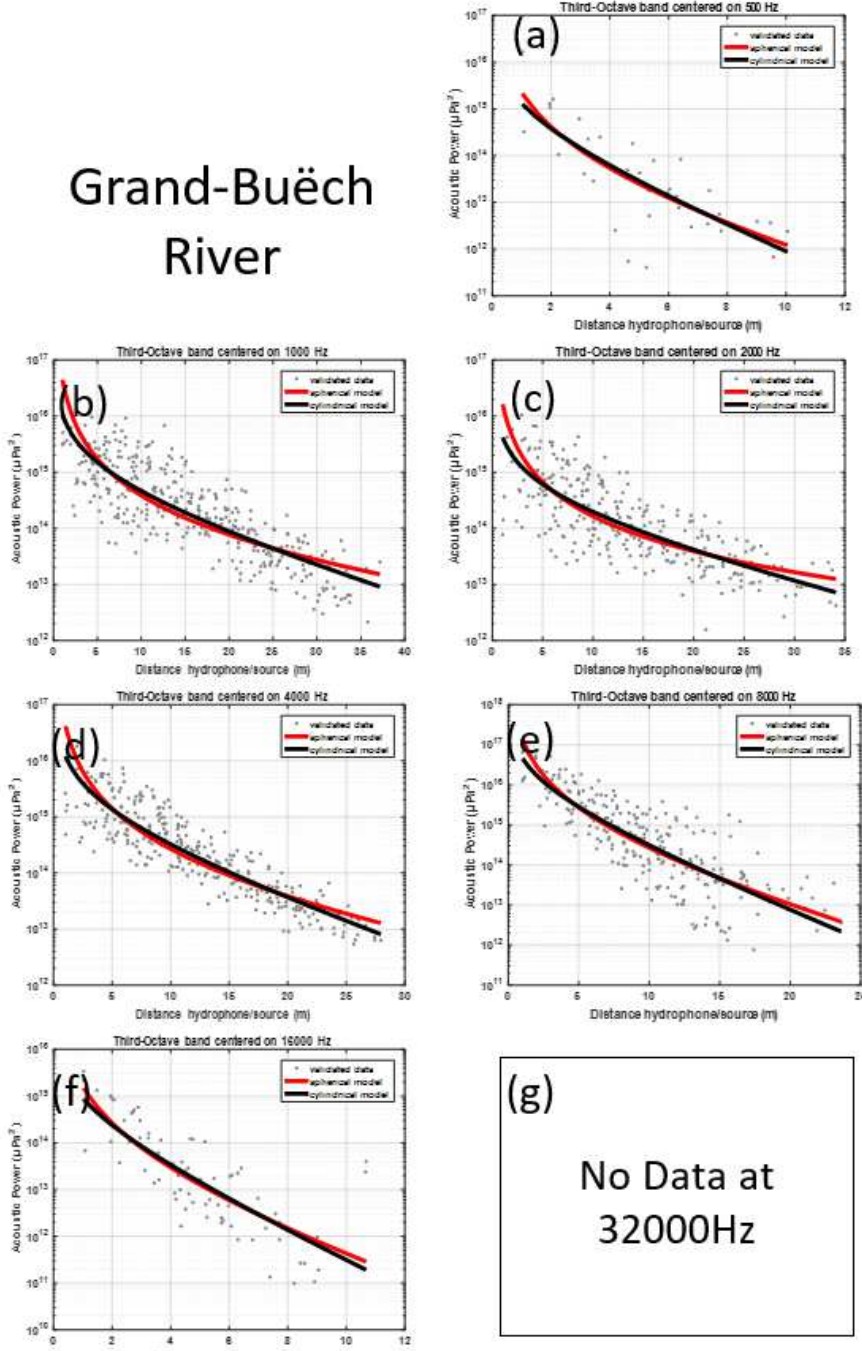

**Figure A2: Data obtained in the Grand-Buëch River. Measured acoustic power (µPa2) in function of the distance between the hydrophone and the active source. Results obtained in third-Octave bands centered on (a) 500Hz, (b) 1000 Hz, (c) 2000 Hz, (d) 4000 Hz, (e) 8000 Hz, (f) 16000 Hz and (g)32000 Hz. Spherical and cylindrical fits are in thick lines.**

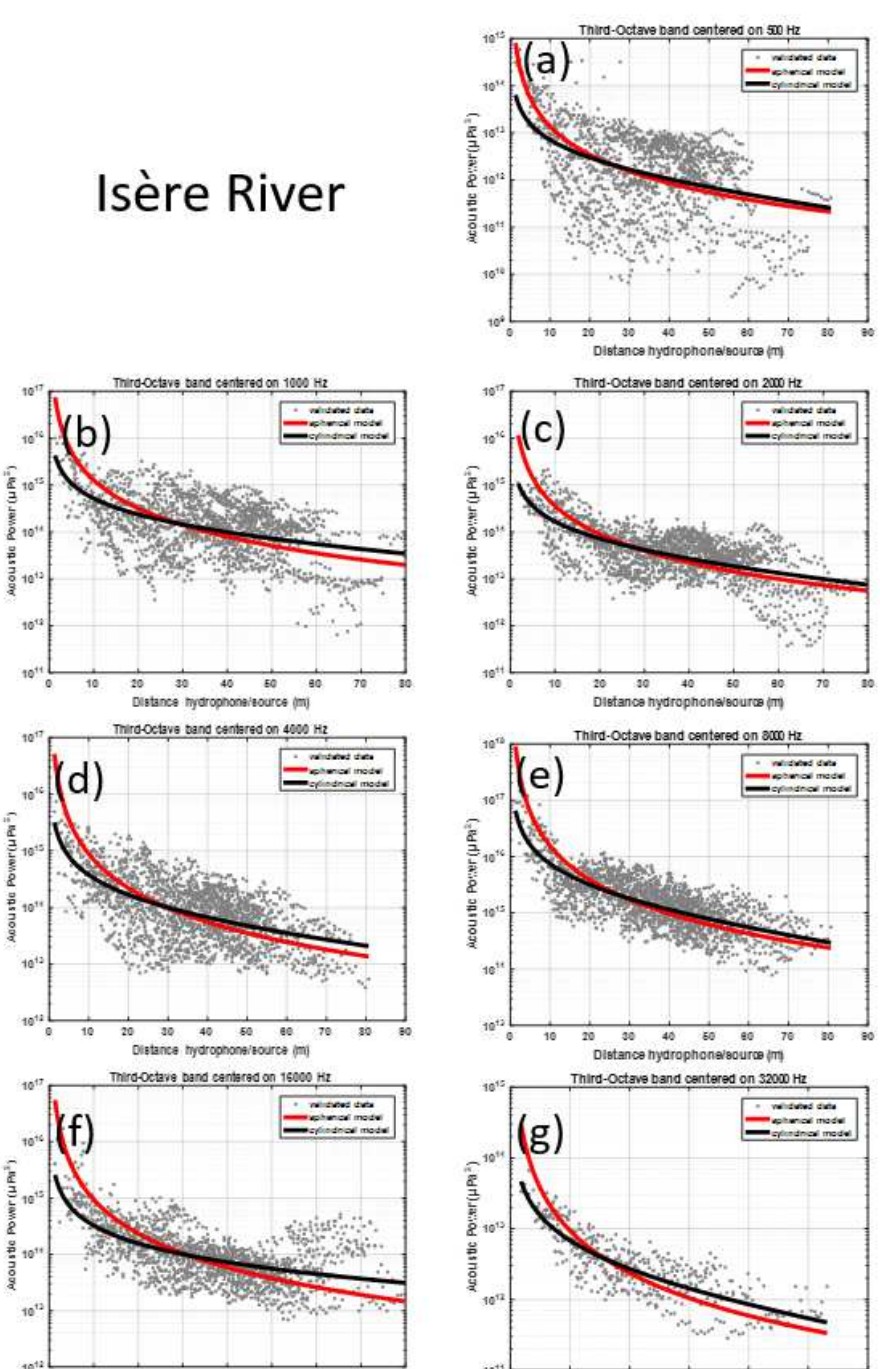

**Figure A3: Data obtained in the Isère River. Measured acoustic power (µPa2) in function of the distance between the hydrophone and the active source. Results obtained in third-Octave bands centered on (a) 500Hz, (b) 1000 Hz, (c) 2000 Hz, (d) 4000 Hz, (e) 8000 Hz, (f) 16000 Hz and (g)32000 Hz. Spherical and cylindrical fits are in thick lines.**

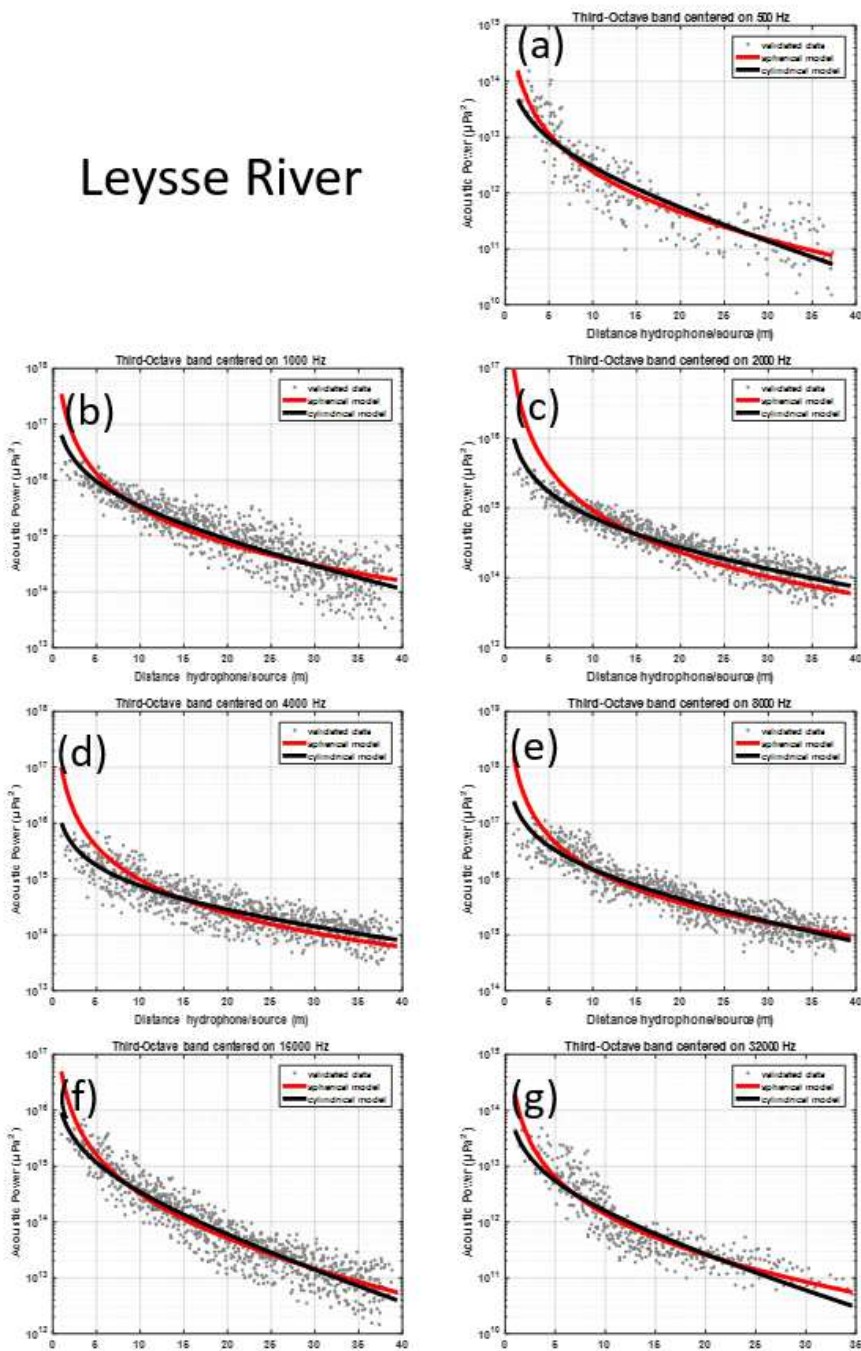

**Figure A4: Data obtained in the Leysse River. Measured acoustic power (µPa2) in function of the distance between the hydrophone and the active source. Results obtained in third-Octave bands centered on (a) 500Hz, (b) 1000 Hz, (c) 2000 Hz, (d) 4000 Hz, (e) 8000 Hz, (f) 16000 Hz and (g)32000 Hz. Spherical and cylindrical fits are in thick lines.**

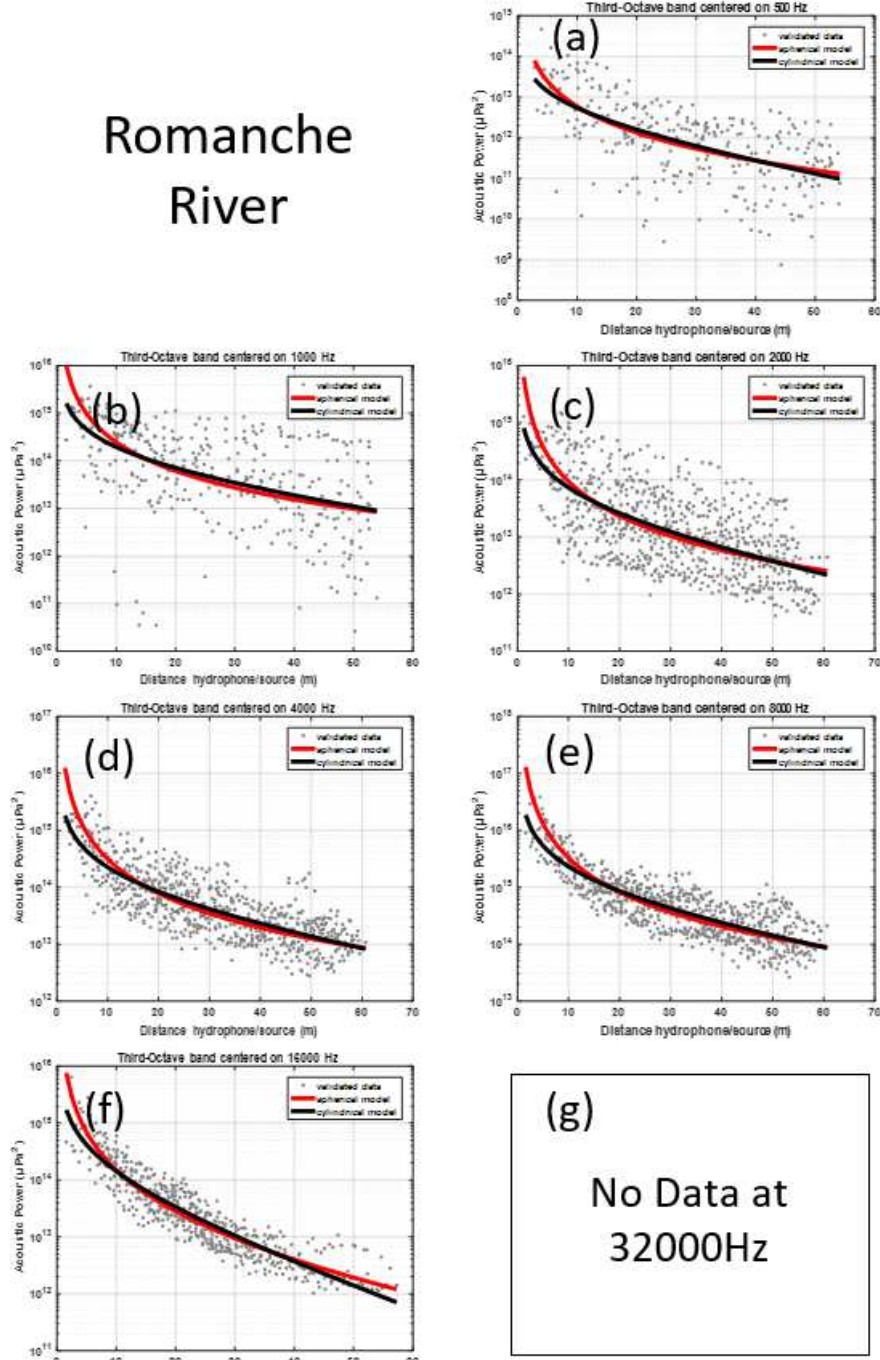

**Figure A5: Data obtained in the Romanche River. Measured acoustic power (µPa2) in function of the distance between the hydrophone and the active source. Results obtained in third-Octave bands centered on (a) 500Hz, (b) 1000 Hz, (c) 2000 Hz, (d) 4000 Hz, (e) 8000 Hz, (f) 16000 Hz and (g)32000 Hz. Spherical and cylindrical fits are in thick lines.**

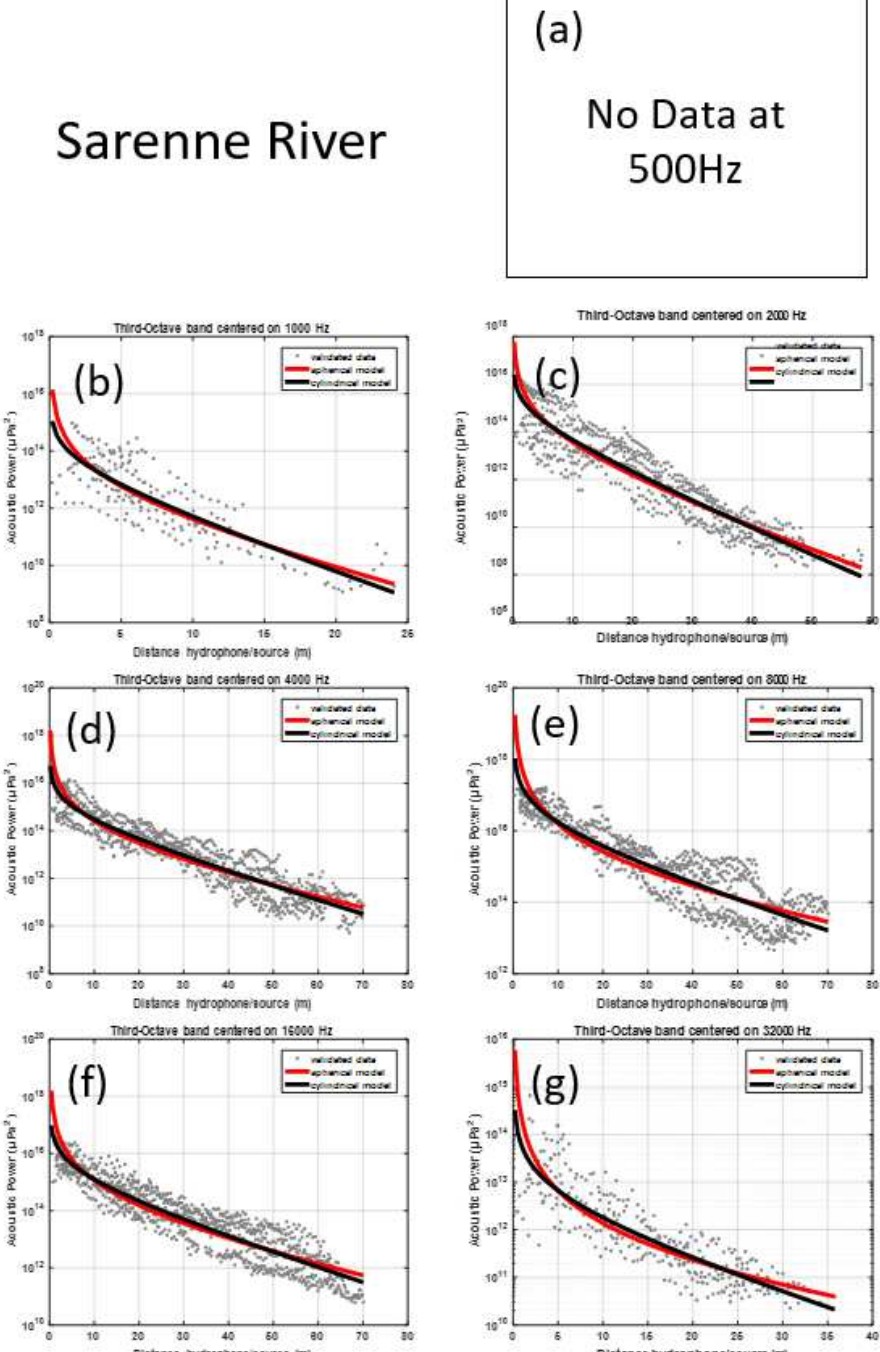

**Figure A6: Data obtained in the Sarenne River. Measured acoustic power (µPa2) in function of the distance between the hydrophone and the active source. Results obtained in third-Octave bands centered on (a) 500Hz, (b) 1000 Hz, (c) 2000 Hz, (d) 4000 Hz, (e) 8000 Hz, (f) 16000 Hz and (g)32000 Hz. Spherical and cylindrical fits are in thick lines.**

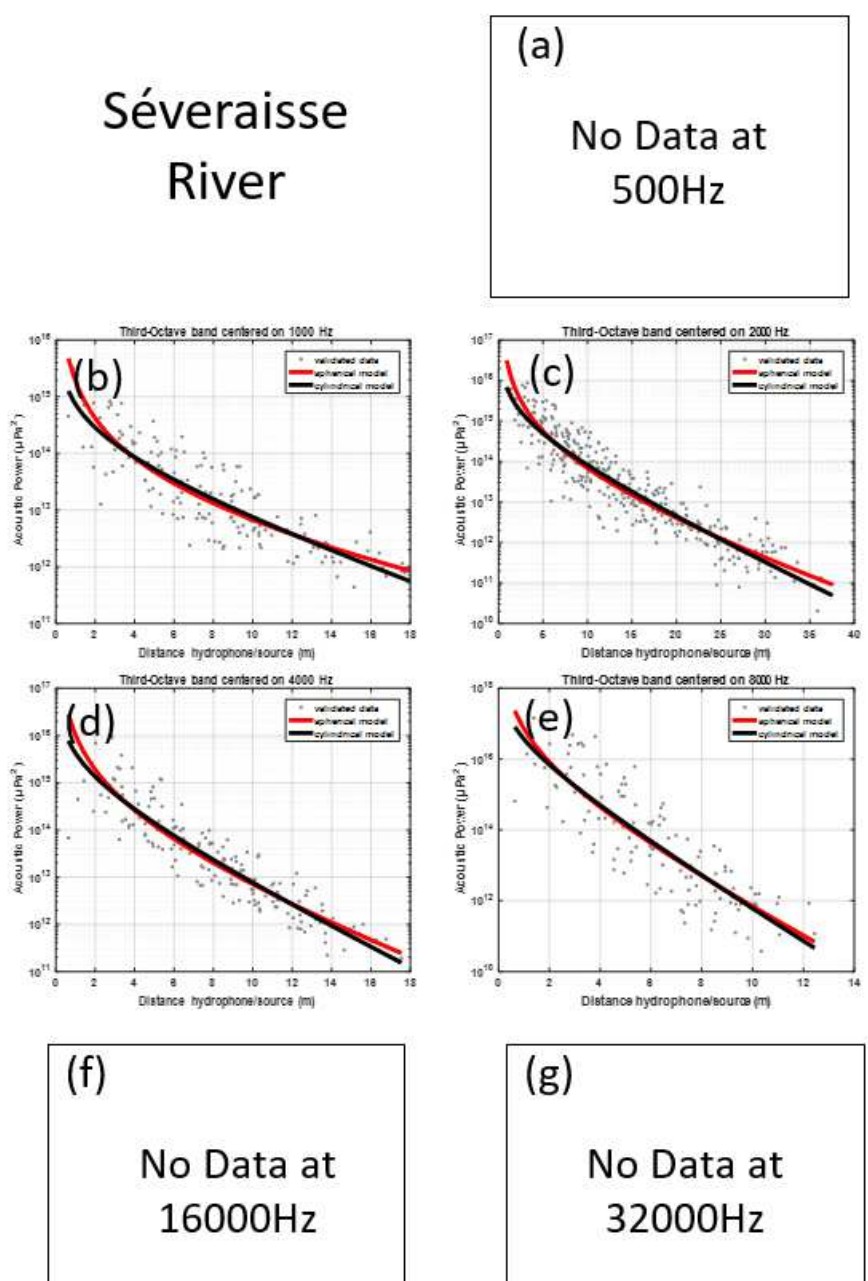

**Figure A7: Data obtained in the Séveraisse River. Measured acoustic power (μPa2) in function of the distance between the hydrophone and the active source. Results obtained in third-Octave bands centered on (a) 500Hz, (b) 1000 Hz, (c) 2000 Hz, (d) 4000 Hz, (e) 8000 Hz, (f) 16000 Hz and (g)32000 Hz. Spherical and cylindrical fits are in thick lines.**

