# Peer review of "Acoustic wave propagation in rivers: an experimental study"

_Earth Surface Dynamics, 2018_

## Referee Comment (RC1) · Anonymous Referee #1 · 25 Jan 2019

Review of 'Acoustic wave propagation in rivers: an experimental study' by Geay et al

General comment.

The manuscript reports on underwater measurements of ambient acoustic noise levels collected in several shallow rivers in the French Alps. The rational for collecting the data is to improve the measurement of the bedload gravel transport, using passive underwater acoustic receivers, hydrophones. Unfortunately, there was an order of magnitude variation in the acoustical noise levels between the different rivers. This variability complicates the generic application of the data to the enhancement of passive acoustic detection of gravel transport. However, given the limited studies of ambient acoustical noise levels in rivers, the data does provide indicative background sound levels, which may be of some value for the passive acoustic detection of gravel transport.

[Figure]

The study may also be of interest to others concerned with acoustical riverine noise levels e.g. naval, marine noise pollution etc. The publication of the work could possibly be considered to be of broader interest than solely the gravel transport community.

Specific comments

1 In the abstract the word 'rugosity' is used, this word is not in common usage; the selection of an alternative to describe this feature of the bed would be helpful.

2 P2 line 5 'frequential characteristics' is a slightly odd phrase, 'spectral characteristics' would be more commonly used.

3 P3 line 11 'interfaces are totally transparent, acoustic waves propagate' it is unlikely that the interfaces would be 'totally transparent', however, they could be 'highly absorbing'.

4 P3 line 20 'c is the celerity of the acoustic waves in water (m/s)'; why not simply say 'c is the velocity of sound in water (m/s)'?

5 On P4 and in figure 2 the transmit sensitivity of the underwater loudspeaker is presented, however, this is only valid if the hydrophones have uniform receive sensitivity over the bandwidth of the transmitter. What was the receiver response over the transmit bandwidth?

6 P4 line 29 It is not clear what is meant by 'shared' in 'The system is shared by a Carlson river board'. Was it 'mounted' on a Carlson river board?

7 P 4 line 30/31 'Lagrangian measurements were preferred to fix-position measurements to optimize the signal to noise ratio.' A few words explaining why this was 'optimize' would be useful.

8 P5 line 12 'describes how are processed the hydrophone signals' 'how the hydrophone signals were processed' would be better.

9 P7 line 5 and fig 5. Some explanation needs to be provided for choosing 1.0 kHz to

assess the acoustic power with range, given that it is cited on P7 line 16 'that estimate a cut-off frequency around1.1 kHz' Why choose to use 1.0 kHz when it is below the cut-off frequency?

10 P7 line 9 'is repeated' should be 'was repeated'.

11 P7 It is not clear in the text how figures 6a and 6b were obtained from the data and how they relate to figure 5. Given this process is central to the manuscript output, it needs to be explicitly and clearly explained. Are the measured spectra in the rivers being scaled to the lake spectra at 1.0 kHz? Are the lake spectral levels being used to obtain the attenuation? Are spectral measurements at different ranges used to calculate the riverine attenuation? Clarification is required if the manuscript is to be published.

12 P7 As with point 11 above it is not clear how the spectra in figure 7 and attenuations in figure 8 were actually obtained from the measurements. Again further clarification is required if the manuscript is to be published. It is not possible to ascertain the veracity of the results presented due to a lack of a clear explanation of the data analysis process.

13 P9 line 2 There needs to be some justification for the choice of 1600 m/s for the sound velocity in the bed sediments.

14 P9 line 9 'The variation of attenuation coefficients at higher frequencies is here discussed' It would be useful to compare the measured attenuations with that calculated solely by the absorption due to the water itself. Was the water absorption a significant component of the measured attenuation in any of the rivers?

15 P10 equation 10. It may be interesting to present equation 10. However, how would the attenuation coefficient be obtained for a new river in which SGN PSD measurements were being collected?

16 P11 line 16 '$\alpha\lambda$ is higher for higher bed slopes of the river'. Any physical explanation

for this?

The manuscript presents a series of observations, which require further explanation as to how the attenuation and source levels are obtained over the spectra presented. In addition, because no ancillary data were collected on the sediments beds and water surface roughness the results presented are of limited value. However, there are not many measurements of riverine soundscapes and therefore it could be considered a publishable manuscript if this is deemed sufficiently original.

Please also note the supplement to this comment:
https://www.earth-surf-dynam-discuss.net/esurf-2018-80/esurf-2018-80-RC1-supplement.pdf

---

## Referee Comment (RC2) · Anonymous Referee #2 · 1 Feb 2019

General comments:

The manuscript describes and discusses an important aspect of a potential new technique for bedload transport measurements in rivers using passive acoustic monitoring with hydrophones. Controlled experiments were performed in seven rivers to assess the sound propagation in stream reaches with site-specific, different morphological characteristics. Using an acoustic source with known characteristics, the attenuation of the sound was determined for different hydrophone positions along the stream channel, essentially determining the cutoff frequency and attenuation coefficients as a function of acoustic frequency. These experiments and the associated findings represent an important step towards a better interpretation and quantification of hydrophone measurements to determine bedload transport in river environments.

[Figure]

Specific comments:

P2L24: These two sentences about results belong rather to the abstract or conclusion section. At this point you should rather more clearly state what the objectives of this study are.

P3L22 and P4L9: The two frequency ranges mentioned are largely similar, but the lower end is different by a factor of 5. You may clarify in section 5.1 why exactly the sound source had a frequency range of 0.2 kHz to 50 kHz.

P4L31 and P7L29: As the hydrophone was fixed at a constant depth from the water surface, it had different relative positions (between water surface and streambed). Although you state in section 3.2 that you did not notice any representative differences in the results for the discharges investigated, you may comment on why different relative positions of the hydrophone may possibly not have a large effect on the results.

P7L14 and P9 top: In the context of eq. (7) you should also indicate the sound speed in water cw (which is only given in the caption of Fig. 8), and discuss the sensitivity of the cutoff frequency fcutoff to uncertainties in the sound speed in the sediment layer cs. For cw = 1450 m/s, h = 1 m, and cs varying from 1500 m/s to 1700 m/s, for example, fcutoff varies by about a factor of 2. What are reasonable bounds for the potential variation of cs?

Fig. 10, Table 1, and Table 2: The values of h/D84 in Fig. 10 are incorrect. I suggest to list these values also in Table 1 explicitly, and to indicate additionally the mean alpha-lambda values in Table 2.

Fig. 10: How was the Froude number determined? Using surface velocity? Using a mean flow depth? Please clarify.

In addition to the important comments no. 11 and no.12 of Referee #1, you should clarify how the mean values of the attenuation coefficients alpha (given in Table 2) and alpha-lambda (given in Fig. 10) were determined (e.g. over which frequency range?).

Technical corrections:

P2L2: Theoretical and experimental studies have shown . . .

P4L16: The Power Spectral Density . . . has been computed

P8L6: the attenuation coefficient varies by more than

P8L27: At "low" frequencies: please give a numeric range of f values here.

P9L5: lithology, grain sizes, porosity . . .

P9L6: but varies from . . .

P9L7: For these reasons, cutoff frequencies are rough estimates and do not . . .

P9L19: Maybe reformulate to: The possible influence of typical nondimensional numbers has also been tested.

P9L27: Also, as observed in a flume experiment . . .

P10L1: difficult to access the riverbed, and . . .

P10L13: and r the horizontal distance from: Do you really mean horizontal or rather bed-parallel, stream-wise direction here?

P10L17: This has several implications for the use . . .

P10L23: measured spectra should be corrected for propagation effects . . .

Fig. 6d: Correct to "(d) Squared correlation coefficient of the fits"

Fig. 6 and Fig. 7: Indicate that measurements refer to the Leysse river (apart from Bourget lake).

Fig. 10c: The abscissa label should read surface D84.

---

## Editor Comment (EC1) · Turowski (Editor) · 4 Feb 2019

Dear authors,

the two reviewers give a host of comments that can be roughly classed in 3 groups: - requests for clarification of statements, methods and other content - language issues - suggestions for the improvement of the analysis and interpretation. Some of these are rather major (e.g., comments 11 and 12, which relate to the description of central analysis for the paper). I ask you, when revising the paper, to think in particular about the reproducibility of your work. Do you provide all necessary information, in terms of methods, assumptions, algorithms etc., that is necessary for a reader to reproduce your analysis? Is all this information organised in an easily accessible way? Is it clearly

[Figure]

ESurfD

Interactive
comment

and unambigiously communicated?

Please provide, in addition to the revised manuscript, a detailed rebuttal describing how you dealt with the comments.

I am looking forward to reading your revised paper,

best wishes, Jens Turowski

---

## Author Comment (AC1) · 28 Feb 2019

Dear Editor and Reviewers, we are pleased to present you a revised version of our manuscript entitled 'Acoustic wave propagation in rivers: an experimental study'. This version has been modified in regards to the detailed comments of two anonymous reviewers. We are grateful to the time given in reviewing our manuscript and to help us in improving our work. A document listing the replies to each review is attached to the revised manuscript (as well as a track-changed version). We hope that you will appreciate this new version and we are looking forward to hearing your opinion. Thomas Geay et Al.

Please also note the supplement to this comment:

[Figure]

https://www.earth-surf-dynam-discuss.net/esurf-2018-80/esurf-2018-80-AC1-supplement.zip

**ESurfD**

---

## Author Response (AR1)

Review of 'Acoustic wave propagation in rivers: an experimental study' by Geay et al

**General comment.**

The manuscript reports on underwater measurements of ambient acoustic noise levels collected in several shallow rivers in the French Alps. The rational for collecting the data is to improve the measurement of the bedload gravel transport, using passive underwater acoustic receivers, hydrophones. Unfortunately, there was an order of magnitude variation in the acoustical noise levels between the different rivers. This variability complicates the generic application of the data to the enhancement of passive acoustic detection of gravel transport. However, given the limited studies of ambient acoustical noise levels in rivers, the data does provide indicative background sound levels, which may be of some value for the passive acoustic detection of gravel transport.

The study may also be of interest to others concerned with acoustical riverine noise levels e.g. naval, marine noise pollution etc. The publication of the work could possibly be considered to be of broader interest than solely the gravel transport community.

| Review                                                                                                                                                                                                                                                                                         | Reply                                                                                                                                                                                                                                                                                                                                                                                                 |
|------------------------------------------------------------------------------------------------------------------------------------------------------------------------------------------------------------------------------------------------------------------------------------------------|-------------------------------------------------------------------------------------------------------------------------------------------------------------------------------------------------------------------------------------------------------------------------------------------------------------------------------------------------------------------------------------------------------|
| 1 In the abstract the word 'rugosity' is used, this
word is not in common usage; the selection of
an alternative to describe this feature of the
bed would be helpful.                                                                                                                | "Rugosity" has been replaced by "Surface grain-
size". "Bed rugosity" has been replaced by "bed
roughness".                                                                                                                                                                                                                                                                                     |
| 2 P2 line 5 'frequential characteristics' is a slightly odd phrase, 'spectral characteristics' would be more commonly used.                                                                                                                                                                    | Done.                                                                                                                                                                                                                                                                                                                                                                                                 |
| 3 P3 line 11 'interfaces are totally transparent,
acoustic waves propagate' it is unlikely that the
interfaces would be 'totally transparent',
however, they could be 'highly absorbing'.                                                                                             | Thank you for the suggestion. We replaced
"totally transparent" by "highly absorbing (as in
an anechoic chamber)"                                                                                                                                                                                                                                                                               |
| 4 P3 line 20 'c is the celerity of the acoustic
waves in water (m/s)'; why not simply say 'c is
the velocity of sound in water (m/s)'?                                                                                                                                                   | Has been changed as suggested.                                                                                                                                                                                                                                                                                                                                                                        |
| 5 On P4 and in figure 2 the transmit sensitivity
of the underwater loudspeaker is presented,
however, this is only valid if the hydrophones
have uniform receive sensitivity over the
bandwidth of the transmitter. What was the
receiver response over the transmit bandwidth? | According to manufacturers, the frequency
response of the loudspeaker is 0.5-21 kHz (+/-
10 dB) and the frequency response of HTI96
hydrophones is 2Hz-30 kHz. AS this is an
important feature, we decided to add this two
sentences:
"The loudspeaker has a frequency response of
+/- 10 dB between 0.5 kHz and 21 kHz, enabling
the generation of sounds in this spectrum." |

**Specific comments**

|                                                                                                                                                                                                                                                                                                    | and
"HTI96 hydrophones have a flat frequency
response between 2 Hz and 30 kHz (+/- 2dB),
enabling absolute measurement of the acoustic
power in this frequency range."                                                                                                                                                                                                                                                                                                                                                                                                                                                                                                                                                                                                                                                                                                                                                                                                                                                                    |
|----------------------------------------------------------------------------------------------------------------------------------------------------------------------------------------------------------------------------------------------------------------------------------------------------|-------------------------------------------------------------------------------------------------------------------------------------------------------------------------------------------------------------------------------------------------------------------------------------------------------------------------------------------------------------------------------------------------------------------------------------------------------------------------------------------------------------------------------------------------------------------------------------------------------------------------------------------------------------------------------------------------------------------------------------------------------------------------------------------------------------------------------------------------------------------------------------------------------------------------------------------------------------------------------------------------------------------------------------------------------|
| 6 P4 line 29 It is not clear what is meant by
'shared' in 'The system is shared by a Carlson
river board'. Was it 'mounted' on a Carlson river
board?                                                                                                                                     | The "system is shared" has been replaced by
"The acoustic recorder and the hydrophone are
shared by"                                                                                                                                                                                                                                                                                                                                                                                                                                                                                                                                                                                                                                                                                                                                                                                                                                                                                                                                            |
| 7 P 4 line 30/31 'Lagrangian measurements
were preferred to fix-position measurements to
optimize the signal to noise ratio.' A few words
explaining why this was 'optimize' would be
useful.                                                                                          | This sentence has been added: "By measuring
when drifting, noises generated by the
resistance of the river board against the flow
are drastically reduced."                                                                                                                                                                                                                                                                                                                                                                                                                                                                                                                                                                                                                                                                                                                                                                                                                                                                                  |
| 8 P5 line 12 'describes how are processed
the hydrophone signals' 'how the
hydrophone signals were processed' would be
better.                                                                                                                                                            | Done.                                                                                                                                                                                                                                                                                                                                                                                                                                                                                                                                                                                                                                                                                                                                                                                                                                                                                                                                                                                                                                                 |
| 9 P7 line 5 and fig 5. Some explanation needs to
be provided for choosing 1.0 kHz to assess the
acoustic power with range, given that it is cited
on P7 line 16 'that estimate a cut-off frequency
around1.1 kHz' Why choose to use 1.0 kHz
when it is below the cut-off frequency? | 1.0 kHz is just an example of the data set for
one frequency band. The paragraph has been
rephrased to read: "As an example, the results
obtained with the third-octave band centered
on 1 kHz are shown in Figure 5."
The cutoff frequency is a rough estimate. The
uncertainty of this estimate has been
highlighted by adding the following sentences:
"The cutoff frequency is dependent on the
water depth (mean water depth of 0.95 m), the
sound speed in water (assumed to be equal to
1500 m/s) and the sound speed in the sediment
layer. Typical values of sound speed in sea floor
materials (from silt to gravel) were observed to
vary between 1550 to 2000 m/s (Jensen et al.,
2011), depending on many factors such as the
type of materials, grain-sizes or porosity
(Hamilton and Bachman, 1982). Using sound
speed of 1550 and 2000 m/s in the sediment
leads to cutoff frequencies of 1500 Hz and
600 Hz, respectively, which is consistent with
our observation." |
| 10 P7 line 9 'is repeated' should be 'was repeated'.                                                                                                                                                                                                                                               | Thanks, done.                                                                                                                                                                                                                                                                                                                                                                                                                                                                                                                                                                                                                                                                                                                                                                                                                                                                                                                                                                                                                                         |

| 11 P7 It is not clear in the text how figures 6a
and 6b were obtained from the data and how
they relate to figure 5. Given this process is
central to the manuscript output, it needs to be
explicitly and clearly explained. Are the
measured spectra in the rivers being scaled to
the lake spectra at 1.0 kHz? Are the lake
spectral levels being used to obtain the
attenuation? Are spectral measurements at
different ranges used to calculate the riverine
attenuation? Clarification is required if the
manuscript is to be published. | An entire sub-section entitled "2.4. Fitting
propagation laws" has been added in the
method section.                                                                                                                                                                                                                                                                                                                                                                                                                                                                                                                                                                                                 |
|---------------------------------------------------------------------------------------------------------------------------------------------------------------------------------------------------------------------------------------------------------------------------------------------------------------------------------------------------------------------------------------------------------------------------------------------------------------------------------------------------------------------------------------------------------------------------------|------------------------------------------------------------------------------------------------------------------------------------------------------------------------------------------------------------------------------------------------------------------------------------------------------------------------------------------------------------------------------------------------------------------------------------------------------------------------------------------------------------------------------------------------------------------------------------------------------------------------------------------------------------------------------------------------------------|
| 12 P7 As with point 11 above it is not clear how
the spectra in figure 7 and attenuations in
figure 8 were actually obtained from the
measurements. Again further clarification is
required if the manuscript is to be published. It
is not possible to ascertain the veracity of the
results presented due to a lack of a clear
explanation of the data analysis process.                                                                                                                                                                                 | An entire sub-section entitled "2.4. Fitting
propagation laws" has been added in the
method section.                                                                                                                                                                                                                                                                                                                                                                                                                                                                                                                                                                                                 |
| 13 P9 line 2 There needs to be some
justification for the choice of 1600 m/s for the
sound velocity in the bed sediments.                                                                                                                                                                                                                                                                                                                                                                                                                                                 | This paragraph has been rephrased: "The cutoff
frequency is dependent on the water depth
(mean water depth of 0.95 m), the sound speed
in water (assumed to be equal to 1500 m/s) and
the sound speed in the sediment layer. Typical
values of sound speed in sea floor materials
(from silt to gravel) were observed to vary
between 1550 to 2000 m/s (Jensen et al., 2011),
depending on many factors such as the type of
materials, grain-sizes or porosity (Hamilton and
Bachman, 1982). Using sound speed of 1550
and 2000 m/s in the sediment leads to cutoff
frequencies of 1500 Hz and 600 Hz,
respectively, which is consistent with our
observation.". |
| 14 P9 line 9 'The variation of attenuation
coefficients at higher frequencies is here
discussed' It would be useful to compare the
measured attenuations with that calculated
solely by the absorption due to the water itself.
Was the water absorption a significant
component of the measured attenuation in any
of the rivers?                                                                                                                                                                                                                         | This sentence has been added: "The attenuation due to freshwater vary from 10 -9 to 10 -3 nepers/m from 1 to 100 kHz (Fisher and Simmons, 1977). The attenuation due to water only do not explain the coefficient of attenuation that were found in this study."                                                                                                                                                                                                                                                                                                                                                                                                                     |
| 15 P10 equation 10. It may be interesting to
present equation 10. However, how would the
attenuation coefficient be obtained for a new
river in which SGN PSD measurements were
being collected?                                                                                                                                                                                                                                                                                                                                                                    | An experimental protocol has been presented
in this paper, it could be used in this new river.
This paragraph has been rephrased as:
"The power generated by bedload sounds is
proportional to the power of measured sounds                                                                                                                                                                                                                                                                                                                                                                                                                                                                    |

|                                                           | multiplied by the attenuation coefficient. [] To                          |
|-----------------------------------------------------------|---------------------------------------------------------------------------|
|                                                           | achieve the estimation of sounds that are                                 |
|                                                           | generated by bedload transport ( PSDs ), both           |
|                                                           | measurements of propagation properties ( $\alpha$ )                       |
|                                                           | and ambient sounds ( PSDh ) are needed. Note            |
|                                                           | that equation 11 was obtained by assuming                                 |
|                                                           | sound sources (i.e. bedload fluxes) that are                              |
|                                                           | homogeneously distributed. As this hypothesis                             |
|                                                           | will rarely be valid, more realistic inverse                              |
|                                                           | methods should be invented to estimate the                                |
|                                                           | real sounds ( PSD s ) generated by bedload              |
|                                                           | transport and its spatial distribution."                                  |
| 16 P11 line 16 ' $\alpha\lambda$ is higher for higher bed | The following sentence has been rephrased to                              |
| slopes of the river'. Any physical explanation for        | read: "It has been found that $\pmb{lpha_{\lambda}}$ was well             |
| this?                                                     | correlated to the slope of the river-bed reaches                          |
|                                                           | (and to the surface D 84 of the emerged bars as         |
|                                                           | well), where $\pmb{lpha}_{\pmb{\lambda}}$ is higher for higher bed slopes |
|                                                           | of the river. Assuming that river-bed slope and                           |
|                                                           | surface D 84 of bars are good proxies for the           |
|                                                           | river-bed texture, it can be concluded that                               |
|                                                           | attenuation properties is dominated by                                    |
|                                                           | processes related to the river-bed roughness at                           |
|                                                           | high frequencies, including the entrainment of                            |
|                                                           | air bubbles in the water column and scattering                            |
|                                                           | effects on rough boundaries."                                             |

The manuscript presents a series of observations, which require further explanation as to how the attenuation and source levels are obtained over the spectra presented. In addition, because no ancillary data were collected on the sediments beds and water surface roughness the results presented are of limited value. However, there are not many measurements of riverine soundscapes and therefore it could be considered a publishable manuscript if this is deemed sufficiently original.

**Anonymous Referee #2**

Received and published: 1 February 2019

General comments:

The manuscript describes and discusses an important aspect of a potential new technique for bedload transport measurements in rivers using passive acoustic monitoring with hydrophones. Controlled experiments were performed in seven rivers to assess the sound propagation in stream reaches with site-specific, different morphological characteristics. Using an acoustic source with known characteristics, the attenuation of the sound was determined for different hydrophone positions along the stream channel, essentially determining the cutoff frequency and attenuation coefficients as a function of acoustic frequency. These experiments and the associated findings represent an important step towards a better interpretation and quantification of hydrophone measurements to determine bedload transport in river environments.

**Specific comments**

| Review                                                                                                                                                                                                                                                                                                                                                                                                                                                             | Response                                                                                                                                                                                                                                                                                                                                                                                                                                                                                                                                                                                                                                                                                                                                                      |
|--------------------------------------------------------------------------------------------------------------------------------------------------------------------------------------------------------------------------------------------------------------------------------------------------------------------------------------------------------------------------------------------------------------------------------------------------------------------|---------------------------------------------------------------------------------------------------------------------------------------------------------------------------------------------------------------------------------------------------------------------------------------------------------------------------------------------------------------------------------------------------------------------------------------------------------------------------------------------------------------------------------------------------------------------------------------------------------------------------------------------------------------------------------------------------------------------------------------------------------------|
| P2L24: These two sentences about results
belong rather to the abstract or conclusion
section. At this point you should rather more
clearly state what the objectives of this study
are.                                                                                                                                                                                                                                                                | Ok, this has been replaced by: "The variation of
propagation properties is observed from one
river to another and related to river
characteristics. "                                                                                                                                                                                                                                                                                                                                                                                                                                                                                                                                                                                                |
| P3L22 and P4L9: The two frequency ranges
mentioned are largely similar, but the lower
end is different by a factor of 5. You may clarify
in section 5.1 why exactly the sound source had
a frequency range of 0.2 kHz to 50 kHz.                                                                                                                                                                                                                       | This sentence has been added:
"The loudspeaker has a frequency response of
+/- 10 dB between 0.5 kHz and 21 kHz, enabling
the generation of sounds in this spectrum."
And this sentence has been precised:
" logarithmic chirp varying from 0.2 kHz to
50 kHz in 1 second, a bit larger than the
theoretical frequency response of the
loudspeaker."                                                                                                                                                                                                                                                                                                                                                                                  |
| P4L31 and P7L29: As the hydrophone was fixed
at a constant depth from the water surface, it
had different relative positions (between water
surface and streambed). Although you state in
section 3.2 that you did not notice any
representative differences in the results for the
discharges investigated, you may comment on
why different relative positions of the
hydrophone may possibly not have a large
effect on the results. | Yes, varying water depth (i.e. varying discharge)
should have an impact on the attenuation
coefficients in the lower frequency range
(because the cutoff frequency is dependent on
the water depth). However, our study did not
experience enough water discharges (levels) to
give significant results. Concerning the effect of
varying relative positions, our hydrophone was
almost set at the same depth for almost same
water levels, and we don't have the data to
show that his effect may have a large or small
effect on the determination of attenuation
coefficients.
In relation to this review: the following
sentence "As we did not notice any
representative differences in the results for the |

|                                                                                                                                                                                                                                                                                                                                                                                                                                                                                               | investigated, we decided to gather data to
propose a unique result for each river." has
been replaced by "For the discharge
investigated, hydrodynamic conditions were
not enough variable to observe major
differences in the results. We therefore
decided to gather data to propose a unique
result for each river. "                                                                                                                                                                                                                                                                                                                                                                                    |
|-----------------------------------------------------------------------------------------------------------------------------------------------------------------------------------------------------------------------------------------------------------------------------------------------------------------------------------------------------------------------------------------------------------------------------------------------------------------------------------------------|----------------------------------------------------------------------------------------------------------------------------------------------------------------------------------------------------------------------------------------------------------------------------------------------------------------------------------------------------------------------------------------------------------------------------------------------------------------------------------------------------------------------------------------------------------------------------------------------------------------------------------------------------------------------------------------------------------------------------------|
|                                                                                                                                                                                                                                                                                                                                                                                                                                                                                               | Secondly, a small paragraph has been added in
the discussion on both aspects (water depth
and relative positions): "Note also that different
hydrodynamic conditions were investigated for
some rivers. Varying water depth results in
different cutoff frequencies and relative
positions of the hydrophone between water
surface and streambed. These two parameters
(water depth and relative positions) have been
observed to modify the response of the
hydrophone (Geay et al., 2017b) in the lower
frequency range, around the cutoff frequency.
The range of experimental conditions that was
investigated in this study did not enable the
characterization of such effects." |
| P7L14 and P9 top: In the context of eq. (7) you
should also indicate the sound speed in water
cw (which is only given in the caption of Fig. 8),
and discuss the sensitivity of the cutoff
frequency fcutoff to uncertainties in the sound
speed in the sediment layer cs. For cw = 1450
m/s, h = 1 m, and cs varying from 1500 m/s to
1700 m/s, for example, fcutoff varies by about a
factor of 2. What are reasonable bounds for
the potential variation of cs? | This paragraph has been rephrased to read:
"The cutoff frequency is dependent on the
water depth (mean water depth of 0.95 m), the
sound speed in water (assumed to be equal to
1500 m/s) and the sound speed in the sediment
layer. Typical values of sound speed in sea floor
materials (from silt to gravel) were observed to
vary between 1550 to 2000 m/s (Jensen et al.,
2011), depending on many factors such as the
type of materials, grain-sizes or porosity
(Hamilton and Bachman, 1982). Using sound
speed of 1550 and 2000 m/s in the sediment
leads to cutoff frequencies of 1500 Hz and
600 Hz, respectively, which is consistent with
our observation."                |
| Fig. 10, Table 1, and Table 2: The values of h/D84 in Fig. 10 are incorrect. I suggest to list these values also in Table 1 explicitly, and to indicate additionally the mean alpha-lambda values in Table 2.                                                                                                                                                                                                                                                                                 | The mean alpha-lambda values have been
added in a new table 3.
Fig. 10 has been corrected (D84 converted from
mm to m). However, the ratio H/D84 is simple
to calculate and as the values of H and D84 are
listed in the table 1, we don't think valuable to
indicate this ratio in the table.                                                                                                                                                                                                                                                                                                                                                                                                                 |

| Fig. 10: How was the Froude number
determined? Using surface velocity? Using a
mean flow depth? Please clarify.                                                                                                                                                        | Previous version was done using surface
velocity, it has been changed using averaged
flow velocity (Q/H*L).
Has been clarified in the legend of fig. 10:
"Froude number computed with averaged flow
velocity and water depth"                                                                                                                                                                                                                                                                                                                                                                                                                |
|------------------------------------------------------------------------------------------------------------------------------------------------------------------------------------------------------------------------------------------------------------------------------|-------------------------------------------------------------------------------------------------------------------------------------------------------------------------------------------------------------------------------------------------------------------------------------------------------------------------------------------------------------------------------------------------------------------------------------------------------------------------------------------------------------------------------------------------------------------------------------------------------------------------------------------------------------|
| In addition to the important comments no. 11
and no.12 of Referee #1, you should clarify how
the mean values of the attenuation coefficients
alpha (given in Table 2) and alpha-lambda
(given in Fig. 10) were determined (e.g. over
which frequency range?). | In the original manuscript, mean values were
determined over the frequency range observed
during the experiment (so variable according to
the field site, see Fig. 8). In the revised
manuscript, a fixed band-width (1-10kHz) has
been used to compute the mean values. This
has been precised in the legend of table 2 and
slight changes can be observed in the values of
table 2.                                                                                                                                                                                                                                               |
|                                                                                                                                                                                                                                                                              | Concerning the mean values of alfa lambda,
they were estimated for different frequency
bands. The lower frequency bound was
determined by looking at the local minimum
observed alfa (alfa function of frequency, figure
8b). The maximum frequency was determined
by the limits of our observations (when
impossible to measure high-pitched sounds at
different distances from the loudspeaker with
too strong attenuation). Finally, to clarify this
aspect of varying frequency bands, an
additional table was added, containing the
limits of the frequency bands over which is
averaged alfa lambda (table 3). |

**Technical corrections:**

| Review                                           | Reply                         |
|--------------------------------------------------|-------------------------------|
| P2L2: Theoretical and experimental studies       | done                          |
| have shown                                       |                               |
|                                                  |                               |
| P4L16: The Power Spectral Density has been       | done                          |
| computed                                         |                               |
|                                                  |                               |
| P8L6: the attenuation coefficient varies by more | done                          |
| than                                             |                               |
|                                                  |                               |
| P8L27: At "low" frequencies: please give a       | "around 1 kHz" has been added |
| numeric range of f values here.                  |                               |
|                                                  |                               |
| P9L5: lithology, grain sizes, porosity           | done                          |
|                                                  |                               |

| P9L6: but varies from                                                                                                             | done                                                                                                                               |
|-----------------------------------------------------------------------------------------------------------------------------------|------------------------------------------------------------------------------------------------------------------------------------|
| P9L7: For these reasons, cutoff frequencies are rough estimates and do not                                                        |                                                                                                                                    |
| P9L19: Maybe reformulate to: The possible influence of typical nondimensional numbers has also been tested.                       | done                                                                                                                               |
| P9L27: Also, as observed in a flume experiment                                                                                    | done                                                                                                                               |
| P10L1: difficult to access the riverbed, and                                                                                      | done                                                                                                                               |
| P10L13: and r the horizontal distance from: Do you really mean horizontal or rather bed-
parallel, stream-wise direction here? | Yes, this is the horizontal distance (assuming
that the horizontal is parallel to the riverbed at
the scale of the section). |
| P10L17: This has several implications for the use                                                                                 | done                                                                                                                               |
| P10L23: measured spectra should be corrected for propagation effects                                                              | done                                                                                                                               |
| Fig. 6d: Correct to "(d) Squared correlation coefficient of the fits"                                                             | done                                                                                                                               |
| Fig. 6 and Fig. 7: Indicate that measurements refer to the Leysse river (apart from Bourget lake).                                | Done for figs. 5 and 6.                                                                                                            |
| Fig. 10c: The abscissa label should read surface D84.                                                                             | Done                                                                                                                               |

[revised manuscript text omitted]
 | Width of    | GSD of                              | Date of field | Water               | mean         | mean     | suspended    |
|------------|-------|-------------|-------------------------------------|---------------|---------------------|--------------|----------|--------------|
|            | slope | the cross-  | emerged                             | experiments   | discharge           | water        | surface  | sediment     |
|            | (%)   | section (m) | bars                                |               | (m 3 /s) | depth        | velocity | concentratio |
|            |       |             | [D 50 -D 84 ] |               |                     | ( m ) | (m/s)    | n (g/L)      |
|            |       |             | ( mm )                       |               |                     |              |          |              |
| Arve       | 0.75  | 14          | [70-120]                            | 2017/06/27    | 38                  | 1.25         | 2.3      | 0.35         |
|            |       |             |                                     | 2017/06/29    | 29                  | 1.1          | 1.95     | -            |
| Grand-     | 0.7   | 13          | [30-66]                             | 2017/04/12    | 5.5                 | 0.35         | 1.5      | < 0.05       |
| Buëch      |       |             |                                     | 2017/05/15    | 12.5                | 0.55         | 1.85     | < 0.05       |
| Isère      | 0.05  | 60          | [23.5-36.5]                         | 2017/03/08    | 171                 | 2.4          | -        | 0.1          |
|            |       |             |                                     | 2017/03/28    | 150                 | 2.3          | -        | 0.06         |
|            |       |             |                                     | 2017/06/06    | 237                 | 2.8          | 1.85     | 0.6          |
| Leysse     | 0.1   | 18          | [39-68]                             | 2017/03/09    | 17                  | 0.95         | 1.2      | < 0.05       |
| Romanche   | 0.13  | 33          | [20-39]                             | 2017/06/14    | 55                  | 1.2          | 1.85     | 0.14         |
| Sarenne    | 0.13  | 8           | [4-8]                               | 2017/04/05    | 1.3                 | 0.3          | 0.7      | < 0.05       |
| Séveraisse | 1.0   | 12.5        | [32-75]                             | 2017/04/25    | 5                   | 0.4          | 1.8      | < 0.05       |

| River       | α                    | Corr. coeff. of           | Residuals     | Maximum distance        |
|-------------|----------------------|---------------------------|---------------|-------------------------|
|             | (nepers/             | the fit (r 2 ) | ( dB ) | of the monitored        |
|             | m)                   |                           |               | chirps (m)              |
| Arve        | 0.2 <mark>67</mark>  | 0.6 5              | 6             | 1 1 2            |
| Grand-Buëch | 0. 08 15      | 0.7                       | 4             | 31 25            |
| Isère       | 0.00 <mark>98</mark> | 0.4                       | 3             | 80 77            |
| Leysse      | 0.0 1 36      | 0.8                       | 2             | 39                      |
| Romanche    | 0.02 <mark>2</mark>  | 0.5                       | 4             | <del>5559</del>  |
| Sarenne     | 0.0 99 82     | 0.8                       | 6 5    | 57                      |
| Séveraisse  | 0.2 15 8      | 0. 8 7             | 5             | 22 <del>19</del> |

Table 2 : Average results over frequency (1-10 kHz) of the parameters of the fit using cylindrical geometrical spreading.

**Table 3:** frequency bands where  $\alpha_{\lambda}$  was observed to be almost constant with frequency and average values of  $\alpha_{\lambda}$  in this frequency range.

| River      | Frequency range | Average αλ |
|-------------------|-----------------|------------------------------|
|                   | [fmin-fmax]     | (nepers/wavelength)          |
|                   | (kHz)    |                              |
| Arve              | [1-13]          | 0.125                        |
| Grand-Buëch       | [1.6-20]        | 0.032                        |
| Isère             | [1-40]          | 0.003                 |
| Leysse            | [2.5-40]        | 0.005                        |
| Romanche          | [2.5-40]        | 0.004                 |
| Sarenne           | [8-40]          | 0.005                        |
| Séveraisse | [2.5-13]        | 0.085                        |

**Figures**

Figure 1: (a) schematic design of the test characterizing the system of emission; (b) photography of the immerged system in the lake 5 of the Bourget (France).

---

## Referee Report (RR1)

Re-review of 'Acoustic wave propagation in rivers: an experimental study' by Geay et al

The manuscript reports on underwater measurements of ambient acoustic noise levels collected in several shallow rivers in the French Alps. The rational for collecting the data is to improve the measurement of bedload gravel transport, by using passive underwater acoustic monitoring, of the sound radiated by inter-particle impacts of mobile material.

The authors have attended to the specific comments from the first review. As mentioned in the previous review all the results flow from the curve fitting to the data illustrated in figure 5. It is unclear whether the example shown in figure 5 for a 1.0 kHz band in the Leysse river is particularly representative of the whole data set. A figure 5 with subplots showing curve fitting for a number of frequencies would provide the reader with additional evidence of the approach. Further plots in an appendix for three of the rivers at different frequencies representing fits to low (Isere), medium (Leysse) and high (Arve) attenuation would be useful. Alternatively, the authors could make the data forming figure 5 available for all rivers and frequency bands as a supplement to the manuscript. This would allow other interested parties carry out similar analysis.

The data provided is of limited value for improving the detection of bedload transport acoustically owing to the substantial variability in riverine soundscapes. However, as mentioned in the previous review, there are limited studies of ambient acoustical noise levels in rivers and these data provide indicative background sound levels. The authors may be interested in other soundscape studies e.g. Vracar, M. S. and Mijic, M. Ambient noise in large rivers, J. Acoust. Soc. Am., 130, 1787–1791, 2011 and any other publications the authors can find for comparison and provide context for their own study.

The above suggestions are left to editorial discretion as to whether acceptance of the manuscript is contingent on the above additions.

---

## Author Response (AR3)

Dear editors, thank you for your review.

1/Appendix has been moved into supplementary materials in a separate file.

2/Secondly, our statement concerning direct sampling has been corrected (p1, l28:l30)

"Additionally, direct sampling cannot provide automatic, continuous measurements during long periods, limited by the storage capacity and the need for operators." Has been replaced by:

"Long term, automatic and continuous measurements of bedload materials has already been achieved with direct sampling (e.g, Turowski and Rickenmann, 2009) but such a monitoring is typically expensive and technically challenging."

Again, thank you for your work.

Thomas Geay